

# The Fire Weather Index Improved for Boreal Peatlands Using Hydrological Modeling and Satellite-Based L-band Microwave Observations

Jonas Mortelmans[1], Anne Felsberg[1], Gabriëlle J. M. De Lannoy[1], Sander Veraverbeke[2], Robert D. Field[3,4], Niels Andela[5], and Michel Bechtold[1]

[1]Department of Earth and Environmental Sciences, KU Leuven, Heverlee, B-3001, Belgium
[2]Faculty of Science,Vrije Universiteit Amsterdam, de Boelelaan 1085, 1081 HV Amsterdam, The Netherlands
[3]Department of Physics and Applied Mathematics, Columbia University, New York, NY, USA
[4]NASA Goddard Institute for Space Studies, New York, NY, USA
[5]BeZero Carbon, London, UK

**Correspondence:** Jonas Mortelmans (jonas.mortelmans@kuleuven.be)

**Abstract.** The Canadian Fire Weather Index (FWI) system, even though originally developed and calibrated for an upland jack pine forest, is used globally to estimate fire danger for any fire environment. However, for some environments, such as peatlands, the applicability of the FWI in its current form, is often questioned. In this study, we replaced the original moisture codes of the FWI with hydrological estimates resulting from the assimilation of satellite-based L-band passive microwave

observations into a peatland-specific land surface model. In a conservative approach that maintains the integrity of the original FWI structure, the distributions of the hydrological estimates were first matched to those of the corresponding original moisture codes before replacement. The resulting adapted FWI, hereafter called PEAT-FWI, was evaluated using fire occurrences over boreal peatlands from 2010 through 2018. Adapting the FWI with model- and satellite-based hydrological information was found to be beneficial to estimate fire danger, especially when replacing the deeper moisture codes of the FWI. For late-season

fires, further adaptations of the fine fuel moisture code show even more improvement due to the fact that late-season fires are more hydrologically driven. The proposed PEAT-FWI should enable improved monitoring of fire risk in boreal peatlands.

## 1   Introduction

Even though boreal peatlands cover only ∼2-3% (4,000,000 km$^2$) of the world's land cover (Xu et al., 2018), it is estimated that they store 600 Gt of carbon (Yu et al., 2010), 25% of the global soil carbon stock, or twice what is stored in the world's

forests (Pan et al., 2011). Under wet conditions, peatlands act as a sink for atmospheric carbon (Gallego-Sala et al., 2018) but due to natural (e.g. climate change or lightning-ignited wildfires) and anthropogenic (e.g. drainage for agriculture and forestry) disturbances, this sink can turn into a source (Turetsky et al., 2002, 2004; Wieder et al., 2009). Since peatlands are often considered to be water-logged most of the year, they are often excluded from wildfire models (Thompson et al., 2019). On the other hand, the effect of wildfires on peatland carbon stocks has been extensively studied (Gray et al., 2021; Kettridge

et al., 2019; Morison et al., 2021; Nelson et al., 2021; Turetsky et al., 2015; Wieder et al., 2009). Furthermore, the link of peat



moisture to peat fire probability has received considerable attention (Lukenbach et al., 2015; Thompson et al., 2019; Wilkinson et al., 2018, 2020). With climate change, boreal peatland fires are expected to become more frequent (Turetsky et al., 2015; Wotton et al., 2010), because the boreal region is expected to experience more severe fire weather (hot and dry conditions) by the end of this century (Flannigan et al., 2016; Wang et al., 2014) and the main source of boreal burned area, i.e. lightning, is

expected to increase (Chen et al., 2021; Flannigan et al., 2013; Krawchuk et al., 2009; Veraverbeke et al., 2017).

Whereas many studies primarily focus on the impact of peat fires (Enayetullah et al., 2022; Gray et al., 2021; Kettridge et al., 2019; Morison et al., 2021; Nelson et al., 2021; Turetsky et al., 2015; Wieder et al., 2009), relatively little attention has been given to improving available approaches of monitoring fire risk originating from drought. The Canadian Forest Fire Weather Index (FWI) system (Van Wagner, 1987) is one of the most well-known systems to estimate fire danger rating based on potential

fire spread, fuel consumption, fire danger, and fire intensity. It relies solely on weather observations such as temperature, relative humidity, precipitation, and wind speed. While originally developed for boreal upland forests, it is now used in almost all other fire environments around the world (de Groot et al., 2015; Di Giuseppe et al., 2020; Taylor and Alexander, 2006). The FWI system is relatively easy to use in various environments because it estimates relative fire danger as a unitless value that is interpreted differently depending on the fire environment (Field, 2020a). It is for example used operationally by the European

Centre for Medium-Range Weather Forecasts (ECMWF) to provide daily forecasts of fire danger (Di Giuseppe et al., 2020).

The FWI empirically combines three moisture codes, each related to the moisture content of a different fuel, based on meteorological information. The fine fuel moisture code (FFMC) represents the moisture content of litter and fine fuels on the surface of a forest stand, where fires usually ignite. The duff moisture code (DMC) and drought code (DC) of the FWI system were originally designed to represent the moisture in loosely compacted decomposing, and deep compact organic matter in

a mineral soil (Van Wagner, 1987). For organic soils like peatlands, DC and DMC are limited for fire danger assessment as they do not strictly represent moisture content in peat layers (Waddington et al., 2012). However, the existence of these moisture codes indicates that the developers recognized the importance of soil moisture in estimating fire danger, but large-scale physically-based hydrological models and satellite estimates were not yet widely available at the time (Krueger et al., 2022). With recent advances in these fields, it is worth rethinking the original definition of these moisture codes, considering the

high volumetric moisture content and different water table dynamics of organic soils compared to mineral soils (Waddington et al., 2012). Waddington et al. (2012) already proposed a simple adjustment to the DC based on water-table model estimates and remote sensing to develop a new peat moisture code.

Several studies have suggested the potential use of satellite-based soil and fuel moisture products to aid in wildfire prediction (Ambadan et al., 2020; Chaparro et al., 2016; Di Giuseppe et al., 2021; Field, 2020b; Forkel et al., 2012; Holgate et al., 2017;

Leblon et al., 2016; Pettinari and Chuvieco, 2020). While some of these studies focus on using remote sensing to estimate the load and moisture content in aboveground fuel (i.e. vegetation), others suggest the use of remote sensing to estimate soil moisture. Ambadan et al. (2020) found that low soil moisture anomalies from the European Space Agency's (ESA) Soil Moisture and Ocean Salinity (SMOS) mission (Kerr et al., 2010) were observed prior to the occurrence of wildfires in the majority of Canadian ecozones. However, the coarse spatial resolution (e.g. 43 km for SMOS; Kerr et al., 2010), the low

temporal resolution (e.g. 2.5 to 3 days; Kerr et al., 2010) and the latency of the current L-band satellite-based products limit





their direct use for daily forecasts at local or regional scales, where data at a higher spatial resolution is necessary (Chaparro et al., 2016). However, via the assimilation of L-band satellite observations into a land surface model (LSM), optimal soil moisture estimates at finer spatial and temporal resolutions can be obtained. For example, the downscaling of brightness temperature (Tb) data from the SMOS (De Lannoy and Reichle, 2016) and the Soil Moisture Active Passive (SMAP) missions is now well established (Reichle et al., 2019). Only recently, these data assimilation frameworks have started to also explicitly account for peatlands (Bechtold et al., 2020; Reichle et al., 2023).

The Canadian Forest Service Fire Danger Group recently published a plan to update the Canadian Forest Fire Danger Rating System, and as part of that, the FWI system. These updates are planned to be published in 2025 and will add new data sources and a more process-based approach. One of the key aspects that will be incorporated is a new peatland moisture code (PMC; CFSFDG, 2021).

In this study, soil moisture estimates of a data assimilation product based on SMOS L-band brightness temperatures and a peatland-specific LSM are used to systematically replace the current FWI moisture codes. The goal is to develop a peatland-specific version of the FWI system (PEAT-FWI), which could then be used operationally in a similar fashion as the current system. We differentiate between early and late-season fires in this study, because we hypothesize that the effect of soil moisture differs between the two. The fuel for early-season fires is mostly the dead plant matter of the previous growing season and thus less influenced by current soil moisture. Previous studies have shown that during spring (early-season), anthropogenic ignitions peak (Parisien et al., 2023; Turetsky et al., 2010). Parisien et al. (2023) found that these anthropogenic ignitions reduce during the vegetation green-up, i.e. the leafing out of deciduous vegetation, and the resulting change in flammability. They showed that the second peak in the fire season in the boreal zone is mostly due to lightning ignitions (Parisien et al., 2023).

For late-season fires, the majority of the fuel in peatland fires is expected to be peat organic material and to a minor extent living vegetation (Davies et al., 2016). Furthermore, the fuel moisture of the latter depends on the peat moisture status (Harris, 2008). We therefore hypothesize that the replacement of the FWI moisture codes will have different effects on estimating early and late-season fire danger.

The paper is structured as follows: Sect. 2 describes the original FWI system, as well as the used data assimilation product and the adaptations to the FWI system. In Sect. 3 and 4 the results of the evaluation of the new FWI system against fire observations are shown and discussed. Lastly, in Sect. 5, the main conclusions of this study are summarized and the outlook for new research is discussed.

## 2 Methodology

### 2.1 The Fire Weather Index

#### 2.1.1 Original FWI

The FWI system is used to estimate the danger of wildfires, i.e. both the chance of ignition to occur and the possible spread of an ignited fire. Figure 1 illustrates the composition of the FWI. It empirically combines four meteorological variables,





**Figure 1.** Structure of the Canadian FWI and adaptations for PEAT-FWI (dashed arrows; EXP1, EXP2, EXP3, and EXP4). With $T_{2m}$ the 2 m air temperature, $RH_{2m}$ the 2 m relative humidity, P the daily precipitation, $V_{10m}$ the 10 m wind speed, t the timestep, and f() representing non-linear, empirical functions.

2 m relative humidity ($RH_{2m}$), 2 m temperature ($T_{2m}$), 10 m wind speed ($V_{10m}$), and daily precipitation (P) into three daily moisture codes, each representing the moisture content of a different fuel type (Van Wagner, 1987). The first moisture code is

the Fine Fuel Moisture Code (FFMC). It represents the moisture content of litter and fine fuels on the surface of a forest stand, where fires usually ignite. It depends on the $RH_{2m}$, $T_{2m}$, P, $V_{10m}$, and the FFMC of the day before. The FFMC is a unitless number that ranges from 0 to 99, with values > 90 being considered extremes. The second moisture code is the Duff Moisture Code (DMC), which depends on $RH_{2m}$, $T_{2m}$, P, and the DMC of the previous day. It describes the moisture content of loosely compacted organic material on the forest floor. It is unitless and while it has no real upper limit, values over 60 are considered

to be extreme. The last moisture code is the Drought Code (DC), representing the moisture content of deep, compact organic soil layers. It is calculated using $T_{2m}$, P, and the DC of the previous day and is, just like the DMC, unitless and open-ended, but values > 400 are considered to be extreme.





The three moisture codes were then used, together with $V_{10m}$, in non-linear empirical functions to compute three behavior indices (Van Wagner, 1987). The first behavior index is the Initial Spread Index (ISI), which is derived from the FFMC and

$V_{10m}$. It represents the ability of a fire to be ignited and spread if the fine fuel is dry, without taking further vertical drying into account. The second behavior index is the Buildup Index (BUI), which is driven by the DMC and DC. The BUI gives an estimate of the amount of dry fuel that is available for a fire. Lastly, the third behavior index, the FWI, is calculated based on the ISI and BUI. The FWI is a measure of fire danger and intensity, before and after ignition, respectively, with higher values representing more favorable conditions for a wildfire to be triggered and grow. For simplicity, hereafter, we only refer

to 'fire danger' in the context of FWI values. However, the FWI system does not provide specific thresholds for different levels of fire danger. Instead, users have to define such thresholds themselves, based on expert knowledge of the local environment (Van Wagner, 1987).

In this study, the meteorological data needed to calculate the weather-based FWI calculations were taken from the NASA Modern-Era Retrospective Analysis for Research Applications version 2 (MERRA-2; Gelaro et al., 2017). For the calculation

of the different moisture codes, the instantaneous values at noon of $RH_{2m}$, $T_{2m}$, and $V_{10m}$ were used for each day. For P, the in situ gauge-corrected accumulated total precipitation of the last 24 h was taken. The actual FWI calculations were done using the code of the Global Fire WEather Database (GFWED; Field et al., 2015) that uses, for historical FWI estimates, MERRA-2 as input weather data. Aside from the four above-mentioned meteorological variables, snow depth (SD) and mean daily air temperature ($T_{avg}$) were also derived from MERRA-2. These two variables set two thresholds to determine whether the FWI

is calculated for a given day and location. If SD $\geq$ 1 cm or $T_{avg} \leq 6$ °C for a certain location on a specific day, the FWI is not calculated, as the occurrence of an ignition is highly unlikely. Additionally, to start the FWI calculations, three consecutive snow-free days are required. In locations where no snow occurs, three consecutive days with a $T_{avg} > 6$ °C are needed. A more detailed description of the use of these thresholds and the start-up of the FWI calculations is given in Field et al. (2015).

### 2.1.2 PEAT-FWI

To adapt the FWI system over peatlands, we propose a new PEAT-FWI system. The key input for the PEAT-FWI comes from a peatland-specific LSM into which L-band passive microwave observations from SMOS (Kerr et al., 2010) were assimilated from 2010 onwards, as described in Bechtold et al. (2020). The peatland-specific LSM is based on NASA's catchment land surface model (CLSM; Koster et al., 2000) and its peatland modules (PEATCLSM; Bechtold et al., 2019). The PEATCLSM model was forced with hourly 0.625° x 0.5° MERRA-2 data and was run at a horizontal resolution of 9 km on the cylindrical

Equal-Area Scalable Earth grid version 2.0 (EASEv2; Brodzik et al., 2012) and a temporal resolution of one day. The domain of the DA system is the same as described in Bechtold et al. (2020), ranging from 170°W 45°N to 95°E 70°N and the peatland distribution in this area for the DA system was taken from De Lannoy et al. (2014). Due to the highly uncertain information on peatland distribution in Eastern Siberia (Xu et al., 2018), this area was excluded from our study domain. If not mentioned otherwise, the term "boreal peatlands" refers to all peatlands in this domain.

For the PEAT-FWI, four experiments were set up, each replacing another part of the original weather-based FWI system, using daily peatland groundwater table and surface moisture content from the peatland-specific data assimilation output. An





overview of the four experiments is given in Table 1 and Figure 1. For the first experiment (EXP1 from here on), DC was replaced by the PEATCLSM groundwater table (zbar). For the second experiment (EXP2), DC was replaced by zbar, just like for EXP1, but additionally, DMC was also replaced by PEATCLSM surface moisture content (sfmc), which represents the

moisture content in the top 5 cm of the soil. The third experiment (EXP3) is similar to EXP2, but also has the FFMC replaced by sfmc. Lastly, the fourth experiment (EXP4) replaces the final FWI directly with PEATCLSM zbar.

Since the range of the zbar (typically -1 to 0 m) and sfmc (0-0.8) is much smaller than the ranges of the different moisture codes of the FWI (e.g. 0-99 for FFMC), a direct replacement of the moisture codes was not possible without massively changing the relative weights of the different moisture codes on the following ISI, BUI, and FWI calculations. Instead, to maintain the

integrity of the original FWI, we matched the temporal Cumulative Density Functions (CDFs) of the PEATCLSM output variables to those of the corresponding moisture codes for each grid cell according to the principle suggested by Li et al. (2010). While this ensures that the CDFs of the PEATCLSM output variables and those of the corresponding moisture codes match, the approach preserves the dynamical features (short-term fluctuations as well as seasonality) of the PEATCLSM output. By doing this CDF-matching separately for each spatial grid cell, the spatial biases in the PEATCLSM output, such as possible

dry biases over the Boreal plains in Canada (Bechtold et al., 2019), were removed.

**Table 1.** Setup of the different experiments showing the moisture code of the FWI that is changed by PEATCLSM output.

| Experiment name | Input from PEATCLSM | Adjusted moisture code |
| --- | --- | --- |
| EXP1 | zbar | DC |
| EXP2 | zbar & sfmc | DC & DMC |
| EXP3 | zbar & sfmc & sfmc | DC & DMC & FFMC |
| EXP4 | zbar | FWI |

## 2.2 Evaluation

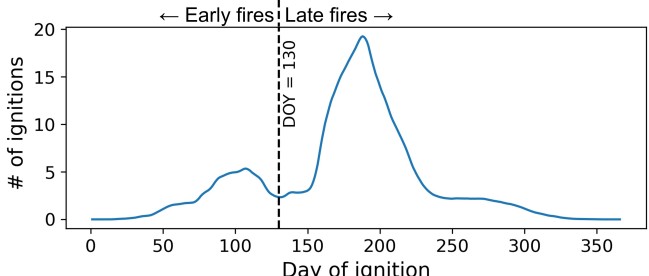

**Figure 2.** Climatology of boreal peat fire occurrence per year based on data from the Global Fire Atlas (Andela et al., 2019) from 2010 through 2018 and the 30 arcsec peatland map. The black dashed line indicates day of year (DOY) 130, which is used to separate the dataset into early and late fires.





The different experiments described in Sect. 2.1.2 were compared against the original, weather-based FWI (FWI$_{ref}$) in their capability to estimate fire danger. As a reference for fire danger, daily peat fire observations were used (Andela et al., 2019). We acknowledge that a high FWI value does not necessarily indicate the presence of a fire, but we assume in the other direction that fire presence indicates high fire danger. Due to the seasonality of boreal wildfires, and to test our hypothesis that late fires are more hydrologically driven, the fires were separated into "early" and "late" season fires, based on the date of the minimum peat fire frequency between the early and late-season fire peak of the multi-year climatology. Figure 2 shows the bimodal climatological histogram of the number of all fires in the boreal zone for each day of year (DOY). The minimum value occurred on the DOY 130, or the 10th of May, which is a bit later than the general start of the fire season (April; De Groot et al., 2013).

### 2.2.1 Peatland map

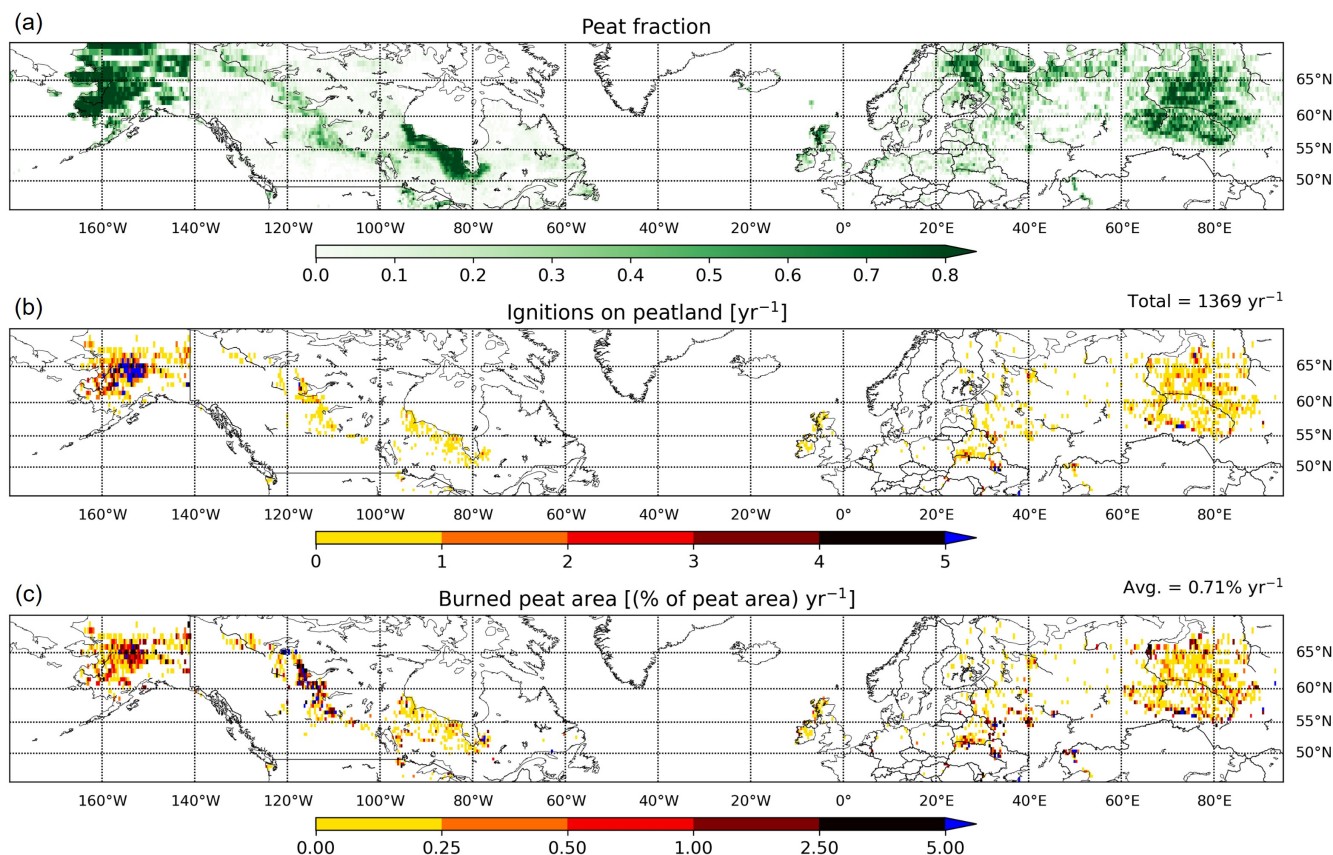

**Figure 3.** Map of the study domain showing (a) the peatland distribution, (b) the annual number of ignitions in peatlands, and (c) the annual burned peat area (as a percentage of peat area in the grid cell) from 2010 through 2018 aggregated to a 36 km EASEv2 grid. The green color bar for the peat fraction indicates which grid cells are considered to be peatland grid cells in this study.





The peatland distribution in this study was based on the one used in the most recent SMAP Level-4 Soil Moisture product (SMAP L4_SM v7; Reichle et al., 2023). This map is a blend of the Harmonized World Soil Database version 1.21 (HWSD1.21; FAO/IIASA/ISRIC/ISSCAS/JRC, 2012), the State Soil Geographic (STATSGO2) dataset (NCRS Soil Survey Staff, 2012) over the United States (including Alaska, Hawaii, and Puerto Rico; De Lannoy et al., 2014), and the PEATMAP (Xu et al., 2018). Due to the relatively low quality of this map over Canada, where PEATMAP only indicates large polygons with an average peatland fraction, we additionally used a new machine-learning based 10-m resolution peatland distribution over Canada, i.e. the Canadian Wetland Inventory Map generation 3 (CWIM3; Mahdianpari et al., 2021), to only select strictly peat locations. The resulting peatland distribution was aggregated to a 30 arcsec resolution and used in the fire data processing described in Sect. 2.2.2. The map is shown at a coarser resolution (36 km) for the Northern Hemisphere in Figure 3 (a).

Since the CWIM3 data is not included in the peatland distribution used for the PEATCLSM simulations, the simulation output was cross-masked with this updated peatland distribution prior to the CDF-matching done in the next steps of this study. The (PEAT-)FWI was thus only calculated for grid cells that are indicated as peatland in both the PEATCLSM simulations and the CWIM3 data.

### 2.2.2 Peat fire dataset

To assess the performance of original FWI and PEAT-FWI, we generated a peat-specific fire dataset from global data on peat fires for the years 2010 through 2018. Peat fire data were derived by combining information from the 30 arcsec peatland map (see Sect. 2.2.1) with the Global Fire Atlas (GFA; Andela et al., 2019).

For the GFA, Andela et al. (2019) grouped burned area pixels based on the Moderate Resolution Imaging Spectroradiometer (MODIS) Collection 6 (Giglio et al., 2018) into individual fire occurrences. From this, they derived the timing and location of ignitions as well as the fire extent. Due to the 500 m spatial resolution of MODIS, very small fires with burned areas $\ll 0.25\,\text{km}^2$ are not contained in the GFA (Giglio et al., 2018).

We then based the differentiation into peat and non-peat fires on the map of peatland fraction at the 30 arcsec resolution ($\sim$500 m). The most intuitive threshold for the minimum peat fraction that indicates the likely presence of a peat fire is 0.5. However, we noted that for the aggregation of the fine spatial resolution of CWIM3 (10 m) to the 30 arcsec binary peat/non-peat grid, the actual peat coverage over Canada would be strongly underestimated using a peat/non-peat threshold of 0.5. This is due to many grid cells having a peat coverage just a little smaller than 0.5. To correct this bias in peatland coverage, and eventually to prevent an underestimation in peat fire occurrence, we adjusted this threshold. First, we derived the percentage of the total Canadian land area that is considered to be peat at the 10 m resolution. Next, different thresholds of peat fraction at the coarse resolution (30 arcsec) were used to determine the fraction that results in the same percentage of the Canadian land area being peat. This threshold was determined to be 0.4, or a 30 arcsec grid cell with at least 40% of the land surface classified as peat is considered a peatland pixel. This threshold is then used for the whole study area but had very little to no effect outside of Canada, as the resolution of the peatland distribution was already coarser originally.

All fires that extended into peatland are defined as "peat fires" independent of their ignition location (outside or inside peatland). Peat fire size is subsequently defined as the peatland area burned within one peat fire and hence typically smaller





than the total fire size, and general statistics about all peat fires based on the 30 arcsec peatland map described in Sect. 2.2.1 are presented in the results.

Next, we focus on the evaluation of the FWI as an indicator of the danger of fire ignition (Sect. 2.2.3 and 2.2.4). For this, we only used the subset of peat fires that were ignited on peatland and for which peat-specific output from PEATCLSM existed

(see cross-masking in Sect. 2.2.1). This means that $FWI_{ref}$ and PEAT-FWI were extracted for the day and location of peat fire ignition alone, even if it is acknowledged that the timing of the MODIS-based fire ignition might contain errors (see Sect. 4.1). If not mentioned otherwise the term "fire" hereafter refers to peat fire.

### 2.2.3 Hits and misses

The first evaluation method, the so-called "Hits and Misses" approach, compares the hits and misses of $FWI_{ref}$ and the PEAT-

FWI. For this, the continuous FWI scale was converted into a binary fire/no-fire scale. The prediction of a fire event is based on a chosen threshold. This threshold was chosen as the 90th percentile of the historical FWI values for the late season, similar to Di Giuseppe et al. (2020), and as the 70th percentile of the historical FWI values for the early season. This threshold is lower for the early season because the FWI values for the early season are much smaller than those of the late season. A hit occurs if an observed fire of the GFA is also predicted to occur with the FWI. If a fire is in the GFA, but the value of the FWI is not

above the threshold, this is classified as a miss.

With this method, the change between the original FWI ($FWI_{ref}$) and the PEAT-FWI of the different experiments can be readily investigated as follows. When $FWI_{ref}$ results in a hit, and the new PEAT-FWI does not, this is noted as a "hit to miss". If it is the other way around, i.e. a miss for $FWI_{ref}$ and a hit for PEAT-FWI, it is called a "miss to hit". To additionally quantify the magnitude of the change from a hit to a miss or vice versa, $\Delta FWI$ is calculated:

$$\Delta FWI = (FWI_{EXP} - Threshold_{EXP}) - (FWI_{ref} - Threshold_{ref}) \tag{1}$$

With $Threshold_X$ the historic 70th percentile of the FWI for the early-season fires, and the historic 90th percentile of the FWI for the late-season fires. Note that $Threshold_{EXP}$ and $Threshold_{ref}$ are not equal after CDF-matching of the soil moisture codes, because they are nonlinearly propagated into the FWI. The somewhat arbitrary threshold levels will be replaced by the full range of thresholds in the next evaluation (Section 2.2.4). Also the effect on false alarms is included in the next section.

### 215 2.2.4 Receiver Operating Characteristics

The receiver operating characteristics (ROC) curve and area under the curve (AUC) were calculated as a second evaluation method. The ROC curve plots the true positive rate (TPR) as a function of the false positive rate (FPR) for varying FWI thresholds. The TPR is the ratio of correctly predicted fire grid cells over all (in space and time) observed fire grid cells. The FPR is the ratio of the erroneously predicted fire grid cells over all (in space and time) observed no-fire grid cells. Because the

predicted fire occurrence depends on a set threshold for the FWI, varying this cut-off value results in a different TPR and FPR. This trade-off between both metrics for different cut-off values is shown by the ROC curve. To indicate the TPR and FPR of the




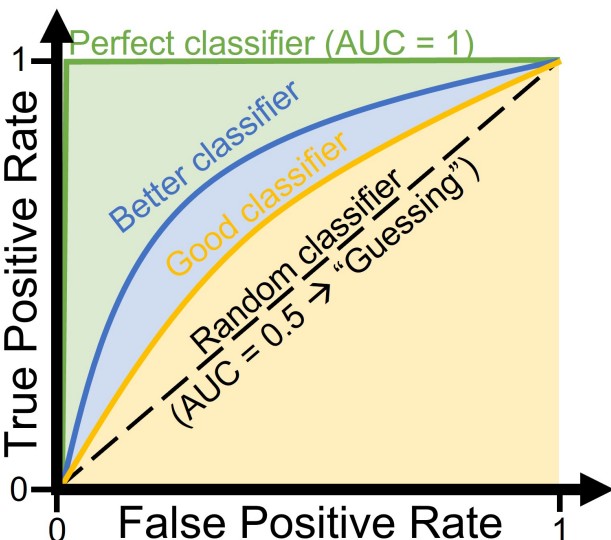

**Figure 4.** Theoretical representation of several receiver operating characteristics (ROC) curves to show classifiers with different predictive capabilities. AUC: Area under the curve.

corresponding threshold used for the "Hits and Misses" approach (the historical 70th (early fires) or 90th (late fires) percentile of the FWI), a star was plotted on the ROC curves.

The ROC curve can be used either to find the optimal cut-off value, i.e. the value for which the point of the ROC is the
closest to the upper left corner or to compare the predictive performance of different models. For the latter, the AUC, a measure of a model's discriminatory power, is used as illustrated in Figure 4. The AUC has a maximum value of 1, indicating a perfect representation of the spatiotemporal fire occurrences by the FWI calculation. A value of 0.5 indicates no discriminative power, i.e. the model does not perform better than a uniform distribution. In other words, the top left corner of the ROC plot is the ideal point, with a TPR of 1 and an FPR of 0. By comparing the ROC curve and AUC of different experiments, their relative
performance can be evaluated. Experiments with a higher AUC, i.e. a ROC curve that goes further to the upper left corner, have a higher predictive capability. Note that the computation of the ROC curve required binary fire occurrence (0: no fire; 1: fire) data, so even if multiple fires were ignited in the same 9 km grid cell on the same day, the fire occurrence in that grid cell is still set to 1.

Lastly, to assess how the (PEAT-)FWI performs for the different regions of the boreal zone, the ROC curve of the late fires
was calculated separately for Alaska, Canada, Europe, and Siberia. Due to an insufficient number of fire events, such a regional stratification was considered unreliable for the subset of early fires.





# 3 Results

## 3.1 Peatland fires

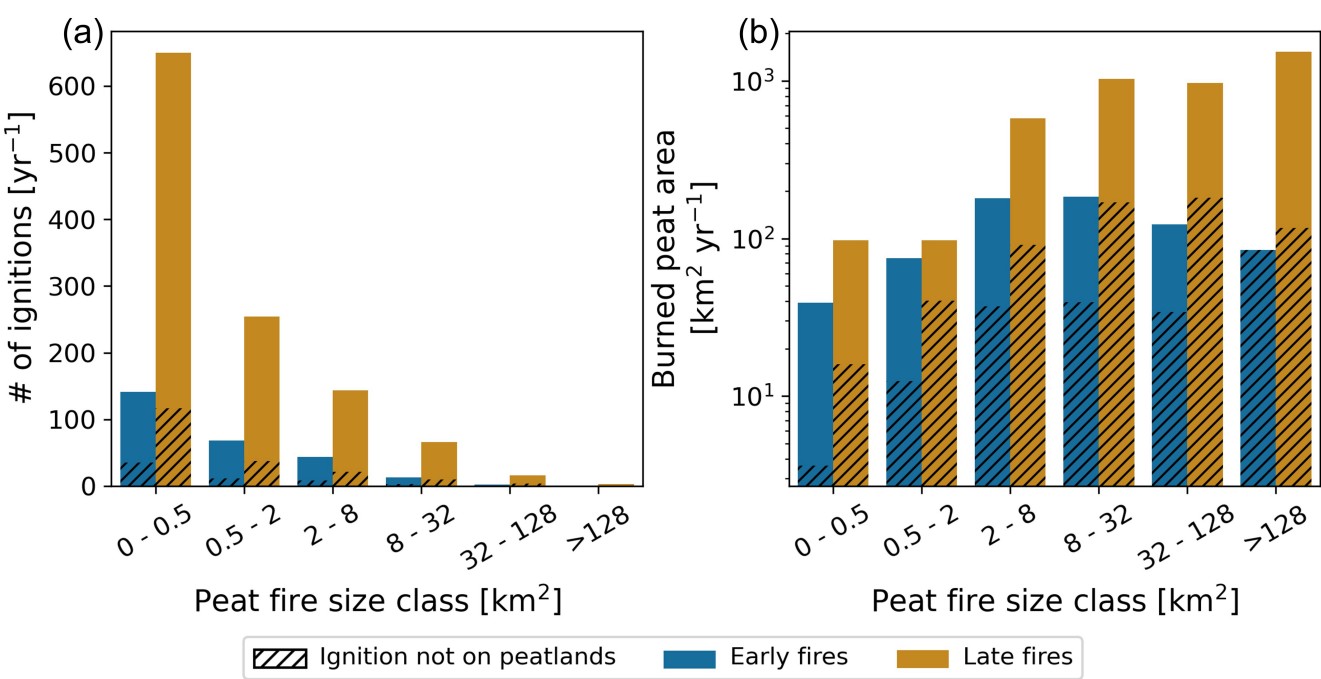

**Figure 5.** Barplots showing (a) the annual number of peat fires and (b) annual burned peat area for different peat fire sizes based on the 30 arcsec peatland map. Wildfires that did not spread into peatlands are not considered here. Black striped parts indicate fires that are ignited outside but moved into peatlands.

Figures 3 and 5 give an overview of the peat fire dataset, described in Sect. 2.2.2, and the peatland distribution based on the

used peatland map described in Sect. 2.2.1, regridded to a 36 km grid for Figure 3, to aid the visualization. The peat fraction shown in Figure 3 (a), shows vast areas of peatland in Alaska, central Canada, the Hudson Bay, Scandinavia, and western Siberia. Figure 3 (b) shows that for most grid cells, there is less than 1 fire per year, but in some of the hot spots, located in central Alaska, this number goes up to almost 30 fires in a 36 km grid cell. However, while central Alaska apparently has much more fires per year, the burned area in central Canada is much larger (see Figure 3 (c)). Note that the burned area here

is represented as the percentage of peat area in the 36 km grid cell (or 1296 km$^2$) that is burned. On average, 0.71% of the peatlands in our study domain burn each year. This means that the average peatland fire return interval for our study domain is approximately 141 years.





Figure 5 (a) shows the number of fires per year for different peat fire sizes, differentiated between early (blue) and late (orange) fires. The fires that burned peatlands but were not ignited on a peatland are indicated in black hatches. In Figure 5 (b),
the same information is shown for the annual burned peat area instead of the number of peat fires.

Comparing the two panels of this figure, and keeping in mind that the y-axis for the right panel is set on a logarithmic scale, it shows that even though the vast majority of peat fires is rather small ($< 2$ km$^2$), most of the burned peat area is caused by relatively few large fires. In fact, $\sim$90% of the burned peat area from 2010 through 2018 is caused by fires $\geq 2$ km$^2$.

### 3.2    PEAT-FWI

Figure 6 shows a time series of the different FWI components for FWI$_{\text{ref}}$ (in blue) and the different PEAT-FWI for a given fire season (June 2012 to September 2012) at a single location (62.68°N, 78.56°W) with multiple fires. By design (see Table 1), some components are the same for different experiments. Only the color of the experiment with the lowest index (e.g. EXP1 and not EXP2) is shown in Figure 6, with an indication of which other experiments have the same time series. The black vertical lines indicate the ignition times of a fire, with the duration visualized by the gray-shaded areas. The black triangle at
the bottom of the FWI time series indicates that this fire was missed by FWI$_{\text{ref}}$, but was a hit for all PEAT-FWI (i.e. a miss to hit for all PEAT-FWI). The gray triangle shows a miss to hit only for EXP3 and EXP4, while this fire is also missed by EXP1 and EXP2. The other fires are either correctly predicted, or missed by all PEAT-FWI and FWI$_{\text{ref}}$.

For all moisture codes (DC, DMC, and FFMC), it can be noted that even though the range of the replaced codes is similar to that of the reference, there are substantial differences in the temporal dynamics. Whenever there is a clear, steep drop in the
reference (blue) time series, there is also (to some extent) a drop in the replaced moisture code. Since CDF-matching does not alter the temporal context of the data but only adjusts the CDF of the PEATCLSM output to that of the FWI$_{\text{ref}}$ components, all differences in dynamics (short- and long-term) between the two datasets are maintained.

In general, over the whole fire season, all the time series of the components of EXP1 and EXP2 show a very similar pattern to those of FWI$_{\text{ref}}$. This is not the case when looking at the replaced FFMC, and consequently ISI, of EXP3. While the FFMC
of FWI$_{\text{ref}}$ shows great day-to-day variation, this is not the case for the FFMC of EXP3. This effect is passed down further through the FWI structure, eventually also resulting in the FWI time series shown in the bottom subplot. When comparing the ISI and FWI of EXP3, one can see that both lines follow a more or less similar pattern. Furthermore, when the FFMC of EXP3 is larger than that of FWI$_{\text{ref}}$, then the ISI and PEAT-FWI are also larger.

By comparing the raw CDF-matched PEATCLSM output (i.e. the DC for EXP1, DMC for EXP2, FFMC for EXP3, and FWI
for EXP4), it is clear that they all still follow the same temporal dynamics. It is only when other, meteorological parameters are introduced (e.g. wind in the ISI calculation for EXP3), that the time series start to deviate. By analyzing the FWI curves in the last subplot of all PEAT-FWI and FWI$_{\text{ref}}$, it can be noted that there is only a minor influence of the DC and DMC on the calculated FWI. For EXP3, in which also the FFMC is replaced, a much larger difference with FWI$_{\text{ref}}$ is seen.





**Figure 6.** Time series of the different FWI components for one location and fire season (June - September) for the different EXP. Note that when multiple experiments have the same modification of an FWI component, that of the lowest experiment index is shown here, with an annotation of the experiments that show the same time series. For EXP4, only the FWI time series is given as zbar is directly used for FWI. Black, vertical lines mark the start of a fire and the gray shadings indicate its duration. The triangles in the bottom plot indicate which fires resulted in a miss to hit for the different PEAT-FWI compared to $FWI_{ref}$. Fires without a triangle at the bottom are either correctly predicted, or missed by all PEAT-FWI and $FWI_{ref}$.





## 3.3 Evaluation

### 3.3.1 Hits and misses

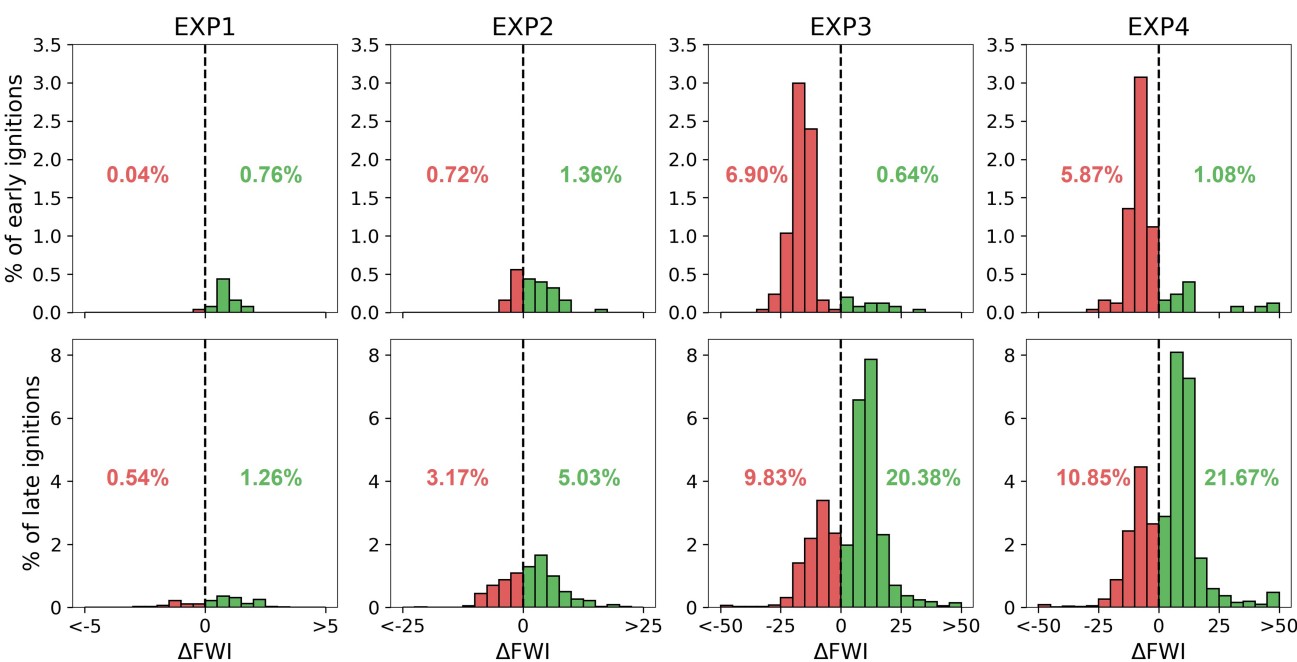

**Figure 7.** Histograms showing the percentage of observed fires (for 2010-2018) that changed from a hit to a miss or vice versa per $\Delta$FWI interval for the different experiments (columns) separated for early (top) and late (bottom) fires. The red bars indicate a "hit to miss" and the green bars a "miss to hit". The percentage indicates the relative number of changes compared to all early ($n = 2506$) or late ($n = 10257$) fires.

Figure 7 shows for each experiment the hits of FWI$_{ref}$ that turned into misses in the different experiments (so-called "hits to misses") in red (negative $\Delta$FWI values) and the misses of FWI$_{ref}$ that turned into hits in the different experiments (so-called "misses to hits") in green (positive $\Delta$FWI values) as the percentage of all observed fires. The top row shows this for the early fires and the bottom row for the late fires. Note that this figure only shows the $\Delta$FWI value if there was a change between

FWI$_{ref}$ and the experiments. If there was a hit or miss in both cases, the $\Delta$FWI value is not shown here. For early fires, there is a clear difference between EXP1 and EXP2 on the one hand, and EXP3 and EXP4 on the other. EXP1 and EXP2 mainly improve the predictive capability of the FWI compared to FWI$_{ref}$ (positive $\Delta$FWI values), while EXP3 and EXP4 show mainly a deterioration of the predictive capability (negative $\Delta$FWI values), even though the absolute number of changed predictions ($n$=20 and 52 for EXP1 and EXP2, respectively) is larger for EXP3 ($n$=189) and EXP4 ($n$=174).

For the late fires, a similar trend in the number of changes can be seen, with $n$ being much larger for EXP3 ($n$=3098) and EXP4 ($n$=3336) compared to EXP1 ($n$=184) and EXP2 ($n$=841). However, this time, EXP3 and EXP4 show much better





results, with approximately 67% of the changed predictions (approximately 20% of all late fires) being a "miss to hit". While EXP1 still has a relatively higher number of "misses to hits", the absolute number of changes ($n$) is much smaller.

This large difference in $n$ between the different experiments indicates that if more of the original FWI structure is changed, the eventual FWI is also changed more. As described in Sect. 2.1.2, EXP1, in which only DC was replaced, is still much more similar to the original $FWI_{ref}$ than e.g. EXP3, where all moisture codes are replaced by PEATCLSM output. Consequently, the calculated FWI remains more similar to $FWI_{ref}$ for EXP1 than for EXP3. That results in fewer changes in the prediction of fires, and thus fewer "hits to misses" and "misses to hits".

### 3.3.2 ROC

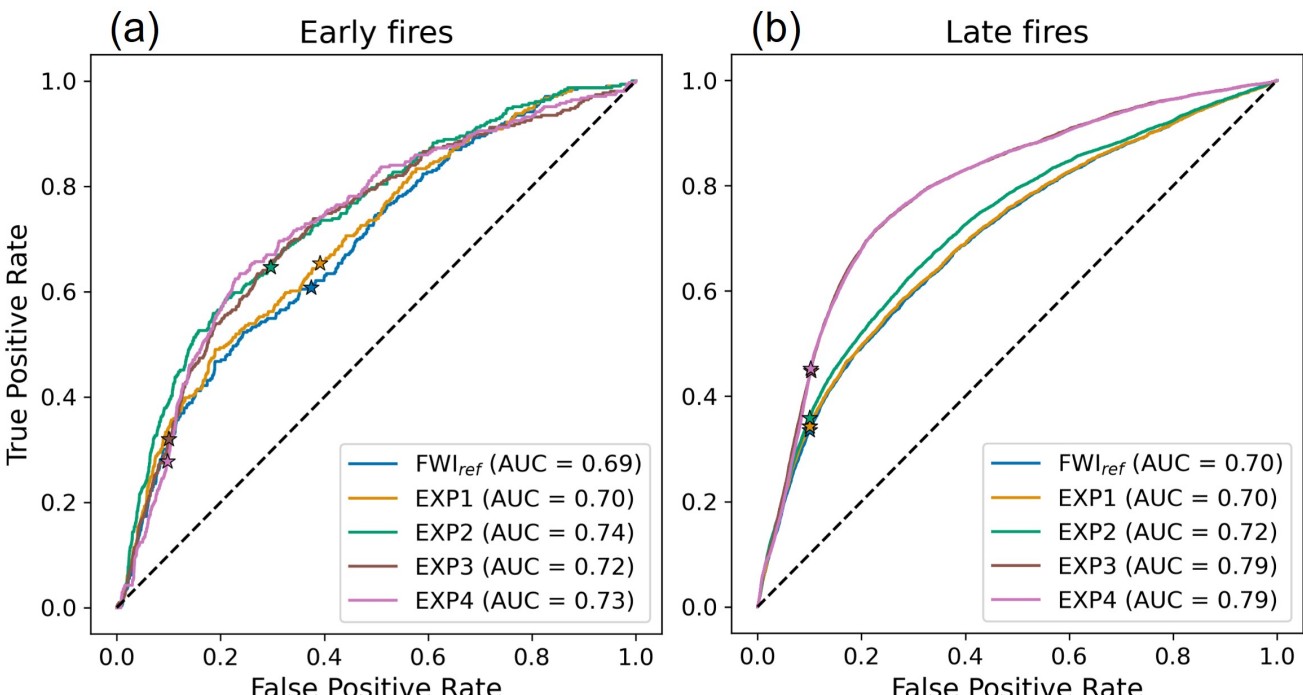

**Figure 8.** ROC curves and corresponding AUC of the different experiments and $FWI_{ref}$ for (a) early ($n = 2506$) and (b) late ($n = 10257$) fires. Stars indicate the true and false positive rate of the 70th (for early fires) or 90th (for late fires) percentile, used as a threshold for the "Hits and Misses" analysis.

Figure 8 shows the ROC curves and corresponding AUC for the early (Figure 8 (a)) and late (Figure 8 (b)) fires for the different experiments. For the early fires, two general trends can be seen. The first trend, followed by $FWI_{ref}$ and EXP1, shows a relatively low AUC (0.69 and 0.70 for $FWI_{ref}$ and EXP1, respectively). EXP2, EXP3, and EXP4 follow the second trend,



with slightly higher AUC values (0.74, 0.72, and 0.73 for EXP2, EXP3, and EXP4, respectively). The stars on the curves indicate the TPR and FPR of the 70th percentile threshold.

For the late fires (Figure 8 (b)), the differences between the experiments are much more pronounced. EXP1 still closely resembles FWI$_\text{ref}$, with both ROC curves having the same AUC (0.70). The AUC of EXP2 (0.72) already indicates a slight improvement, but the difference is still minor. The biggest improvements are seen for EXP3 and EXP4 (AUC = 0.79 for both EXP). The stars on this figure indicate the TPR and FPR of the 90th percentile threshold. Compared to the early fires, the stars here are much more aligned, i.e. they have a similar FPR, but varying TPR, for a set threshold.

Figure 9 shows the ROC curve and corresponding AUC values for late fires in Alaska, Canada, Europe, and Siberia. Only the late fires are considered here, as this covers the main fire season and not all regions had sufficient early fires. All regions show similar results as Figure 8 but the magnitude of the improvement differs between regions. The largest improvement can be seen for Alaska, with an AUC increase of 0.12 for EXP3 and EXP4 compared to FWI$_\text{ref}$. The smallest improvement can be seen for Europe (AUC increase of 0.06 for EXP4 compared to FWI$_\text{ref}$) but notice that all AUC values of Europe are already higher than any of the AUC values of the other regions. While Europe shows the best performance of all PEAT-FWI and FWI$_\text{ref}$, Canada

shows the worst performance, with AUC values ranging from 0.67 for FWI$_\text{ref}$ to 0.75 for EXP3.

## 4 Discussion

### 4.1 Peat fire dataset

For the evaluation of the predictive capabilities of the FWI, fire observations from 2010 through 2018 of the GFA are used, as

discussed in Sect. 2.2.2. Figure 3 shows an overview of this dataset. The average fire return interval of approximately 141 years found in this study is hard to compare with other studies, due to the study domain, time period, and the focus on peatlands alone. However, these results are similar to the results of Wilkinson et al. (2023), who found a spatially weighted average fire return interval of 290 years for boreal and temperate peatlands (Wilkinson et al., 2023). Other studies that quantify fires in (part of) the boreal region, usually do not differentiate between upland forest fires and peatland fires. However, such studies can still

give an indication as to whether these numbers are in line with previous estimates or not. Flannigan et al. (2016), for example, showed that in Canada there are on average 8000 fires per year, burning a total area of more than 2 million hectares. Since they only looked at fires in Canada and did not differentiate between peatland and non-peatland fires, these numbers are in the same range as the fires of the GFA used in this study when taking into account the peatland fraction over Canada (∼12%; Xu et al., 2018). Turetsky et al. (2015) did a global peatland study to investigate peat fire vulnerability. They showed that, for the boreal

region, there is an average fire return interval between 100 and 1000 years which encompasses the average value found in our study. Turetsky et al. (2010) found that, over the full boreal domain, 12 million hectares (120,000 km$^2$) burned each year from 2000 until 2010. This number is not peatland-specific but rather entails all land surface types and ∼20 times larger than the value we found for our study domain, including boreal peatland fires alone.

    There are some shortcomings of the GFA dataset that are important to mention. The GFA dataset is based on MODIS (Giglio

et al., 2018) observations to create individual fire occurrences (Andela et al., 2019). The first, and most obvious shortcoming





**Figure 9.** ROC curves and corresponding area under the curve (AUC) of the different experiments and $FWI_{ref}$ for late fires, separated for Alaska ($n = 6091$), Canada ($n = 849$), Europe ($n = 1366$), and Siberia ($n = 1951$). Stars indicate the true and false positive rate of the 90th percentile, used as a threshold for the "Hits and Misses" analysis.





is the spatial resolution of MODIS of 500 m. Because of this, fires < 0.25 km² are not observed and thus not registered in the GFA, creating a bias towards the prediction of larger fires (Andela et al., 2019).

The second, and perhaps the biggest, shortcoming of the GFA over the boreal region is the fragmentation of fire events. Xu et al. (2022) state that due to omission errors in the remote sensing products of burned area, fragmented burned pixels originating from the same fire can be seen as separate ignitions, likely resulting in an overestimation of the number of ignitions in the GFA.

The third shortcoming is that the dating of ignitions could be erroneous with estimates of ignition dates being either too early or too late. Andela et al. (2019) stated that the burn dates from the GFA mostly corresponded to those observed by active fire detections, but a significant fraction of fires was detected either one day earlier or later. In some cases, e.g. following persistent cloud cover, it is possible that burned area is detected days or weeks after the actual fire event occurred. Since the uncertainty of ignition dates in the GFA is not known, we could not take this into account in this study. However, this error can have a large impact on the performance of the PEAT-FWI and FWI$_{\text{ref}}$. To investigate the effect of this uncertainty, additional analyses are needed, e.g. using other fire observations, e.g. the Visible Infrared Imaging Radiometer Suite (VIIRS) active fire product (Schroeder et al., 2014), or by allowing a range of possible FWI sample dates. Besides satellite observations, operational fire data, such as the national burned area composite for Canada (Hall et al., 2020), could be used. Such a further investigation could help to confirm that the PEAT-FWI is superior over FWI$_{\text{ref}}$.

Apart from the GFA, also the peatland map that was used to separate peat from non-peat fires has some shortcomings. The map is a blend of different, partly country-specific, peatland maps. Definitions of what can be classified as a peatland as well as past mapping efforts can substantially vary across countries and datasets (Xu et al., 2018; Reichle et al., 2023; Mahdianpari et al., 2021). As a consequence, the resulting peatland fraction can be regionally biased or characterized by higher uncertainties and also show discontinuities in peatland fraction at country borders. One of the most pronounced discontinuities can be seen between Canada and Alaska which is likely indicative of either an overestimation of peatland fraction in Alaska or an underestimation in Canada, or both. Those biases may have propagated into the number of ignitions and the burned area shown in Figure 3.

## 4.2 Impact of adjusted components on PEAT-FWI

From the time series of the PEAT-FWI values alongside the reference FWI$_{\text{ref}}$ (Figure 6, it becomes apparent that as the original FWI structure is increasingly altered, the differences between the corresponding PEAT-FWI and FWI$_{\text{ref}}$ also become more pronounced. For the DC and DMC of both FWI$_{\text{ref}}$ and the different PEAT-FWI, a generally linear increase is seen during the first month. As these two values reach their maximum, five fires are observed, indicating that they estimate fire danger relatively well. Note that the FFMC is almost always high during this period, with some dips mainly at the end, when there are no fires observed. While the DC and DMC of the different experiments generally follow a similar trend as the DC and DMC of FWI$_{\text{ref}}$, this is not really seen for the FFMC. This similar general pattern for DC and DMC is as expected, because the PEATCLSM output variables are CDF-matched to the moisture codes, making them by definition very similar. The remaining differences can be explained by the differences in temporal patterns between PEATCLSM and the original moisture codes.





While the temporal CDF-matching ensures that the frequency distribution of the PEATCLSM output variables matches that of the original FWI moisture codes, both the short-term fluctuations and seasonality of the PEATCLSM variables are still maintained.

     Contrary to the DC and DMC, the FFMC of $FWI_{ref}$ shows a lot of day-to-day variation. An explanation for this can be found in the way these moisture codes are designed and what they represent. Van Wagner (1987) designed the FFMC to represent

the moisture content in the litter layer and described this layer as relatively fast-drying. Being the top layer of the whole FWI structure, the FFMC is most reactive to rainfall. The DMC and DC, on the other hand, representing roughly the top 5-10 cm and 10-20 cm layers of the soil (De Groot, 1987), are much less sensitive to the short-term rainfall dynamics due to high storage capacity. The top layer of the system will react faster to a rainfall event than the deeper layers, as the precipitation reaches this layer before it reaches the deeper layers. Additionally, the FFMC, being the top layer, is also more sensitive to relative

humidity, wind, temperature, drying, and smaller rainfall events than the deeper moisture codes, as a small rainfall event might not reach these deeper layers. Van Wagner (1987) represented the difference in sensitivity to drying as the time lag, or the time necessary to lose approximately two-thirds of the free moisture above equilibrium. They showed that this time lag for FFMC is two-thirds of a day, for DMC 12 days, and for DC 52 days (Van Wagner, 1987). This shows that FFMC requires much less time to dry out, and thus increases again more rapidly after a rainfall event, resulting in the very spiky pattern that we see in

Figure 6.

     For the replaced FFMC, this spiky pattern is not seen. This can be explained by the hydrology of peatlands and the way this is represented in PEATCLSM. Peatlands typically have many macropores in their surface layers (Bechtold et al., 2019). These macropores allow water to quickly infiltrate without saturating the surface layer. Furthermore, the high hydraulic conductivity of peat surface layers even under unsaturated conditions maintains a strong capillary link between the water table and surface

moisture content during evaporation periods (Bechtold et al., 2018). As a consequence, peat moisture in surface layers is not as reactive to precipitation and drought periods as this is observed for mineral soils. This strong vertical coupling that characterizes peatlands is not represented in the original FFMC, resulting in the spiky pattern that is shown in Figure 6. By replacing the FFMC with PEATCLSM sfmc, this strong vertical soil moisture coupling is introduced into the FWI system, resulting in a smoother FFMC.

The biggest difference in FWI compared to the $FWI_{ref}$ is seen for EXP3 and EXP4. For EXP3, this is a consequence of the additional replacement of the FFMC, as compared to EXP2, indicating the importance of FFMC in the FWI calculations and in the estimation of fire danger. This importance of the FFMC corresponds to findings in other papers. Van Wagner (1987) already stated that the FFMC relates best to fire occurrence. Chaparro et al. (2016) showed that, as FFMC determines the fuels' flammability, it is one of the most important variables to predict fire occurrence in Spain. Also De Jong et al. (2016) showed

that the FFMC, ISI, and FWI are the most important fire predictors in bogs in the UK. However, De Groot and Flannigan (2014) and Wotton (2009) specify that the FFMC is especially a good indicator of fire occurrence in the case of human-induced fires. For lightning-ignited fires, Wotton (2009) found that the DMC is the primary predictor.

     For EXP4, only the FWI is shown in Figure 6 (pink), as the other components are not used or calculated for this experiment. In general, the FWI of EXP4 shows a similar trend as the FWI of EXP3 but shows hardly any short-term fluctuations. This



shows the impact of wind on the daily FWI calculations. A windy day effectively increases the FWI value, compared to a less windy day with the same $RH_{2m}$, $T_{2m}$, and P.

### 4.2.1  Different impacts for early and late fires

Figure 7 and Figure 8 show, as expected, that changes in FWI increase with increasing number of adapted moisture codes. Since the changes in the moisture codes do not propagate linearly through the FWI structure into the final FWI value, there are

some clear differences in how the FWI changes when replacing different moisture codes. In EXP1, only the DC is replaced by the PEATCLSM zbar, resulting in a minor difference between this PEAT-FWI and $FWI_{ref}$ for both the "Hits and Misses" and the ROC analysis. In EXP2, when not only DC is replaced but also the DMC with PEATCLSM sfmc, a slightly larger change can be seen, especially for the late fires. The amount of change between EXP1 and EXP2 is however not linear. This can be explained by the limited and variable weight the DC actually has in the BUI calculation when compared to the DMC.

Van Wagner (1987) developed the BUI to mainly depend on the DMC, especially as it approaches zero. They explicitly stated that whenever the DMC is zero, the BUI is also zero, regardless of how high the DC is. Then, as the DMC rises, the weight of DC also rises until DMC and DC are equal (Van Wagner, 1987). This varying weight of DC results in a highly nonlinear change of FWI with a change in DC. This overall small weight of DC also explains why there is not much improvement, or change in general when comparing the results of EXP1 with $FWI_{ref}$. The generally larger weight of the DMC is shown in the

larger improvements of EXP2. Especially for the late fires, this influence can be seen.

The much bigger changes (positive and negative) for EXP3 compared to EXP2, show the importance of the FFMC in the FWI structure, especially for the estimation of fire danger. This is in line with how Van Wagner (1987) designed the FWI structure. They state that the FFMC, being the indicator of fine surface fuel moisture content, is most related to fire occurrence. This can be explained by the fact that fine fuel will be the first fuel type to catch fire in case of ignition. Of course, one still

needs an ignition for a fire to start. The fact that EXP3 mainly shows misses to hits and even fewer hits to misses than EXP1 and EXP2 for the early fires indicates that these early fires are not so much driven by hydrology. For EXP4, in which the FWI is directly replaced by PEATCLSM zbar, the biggest change is seen, both for the ROC and "Hits and Misses" analysis, especially for the late fires. This, together with the results of EXP3 for the late fires, strongly indicates that the late fires are indeed more hydrologically driven, supporting the hypothesis.

Based on the results of this evaluation with differentiation into early and late fires, one could argue for a variable PEAT-FWI based on the day of the year. For early fires, only replacing the DC and DMC (EXP2), seems to be most suited. While this PEAT-FWI does not show the most changes compared to the other PEAT-FWI, it does show the most improvements which is generally preferred. On the other hand, additionally replacing FFMC (EXP3), or directly replacing the FWI with PEATCLSM zbar (EXP4), shows the most potential for late fires. Further research is needed to investigate in more detail what the optimal

day of the year is to switch from one method to the other which may vary by region. Another option might be to introduce a transition period between early and late fires over which one gradually changes from EXP2 to EXP3 (or EXP4) FWI estimates. However, additional research is needed to find an optimal transition between our recommended FWI adjustments for early and late-season fires.





The regional stratification of the ROC analysis shows that, overall, the positive impact of including peat-specific hydrological
variables in the FWI is consistent across all four regions considered in this study, as shown in Figure 9. While the magnitude
of improvement differs between the regions, the same conclusions can be drawn for each region individually as for all regions
combined. EXP3 and EXP4 show for all regions the largest improvement for the late fires, while EXP1 shows minor to no
improvements compared to FWI$_{ref}$. Overall, the PEAT-FWI performs worst over Canada, which could be related to the more
aggressive fire management compared to e.g. Siberia, which influences the fire behavior overall. It can cause fires with a very
high PEAT-FWI to be extinguished before being detected by satellite remote sensing, eventually lowering the performance of
the PEAT-FWI.

All AUC values are generally higher for Europe and Siberia than for Alaska and Canada, which could be related to the
difference in fire regimes between North America and Eurasia. De Groot et al. (2013) found that Canada had fewer, but more
intense fires, i.e. more severe fire weather, than Russia. They also found that the Canadian fire season peaks earlier than the
Russian fire season (April-May for Canada versus May-June for Russia; De Groot et al., 2013). This could explain part of the
the difference seen here between these regions. As the fire occurrence in Canada peaks in April-May, part of this peak is
considered in the early fires and part in the late fires, in this study. However, the differences seen here are most likely a result of
a combination of various factors. To gain a comprehensive understanding, further research exploring the full range of factors
contributing to these differences is needed.

The performance assessment of any purely fuel-based fire danger estimation is influenced by the dynamics in ignitions (i.e.
lightning and anthropogenic ignitions) that are by design not taken into acount, leading to possible missed events. Similarly,
management measures are not taken into account, making the fire danger estimates vulnerable to false alarms (Di Giuseppe
et al., 2020). Parisien et al. (2023) showed that in spring, before the greenup of vegetation, most ignitions are anthropogenic.
On the other hand, in summer, lightning is the main ignition source. Since anthropogenic ignitions are much less dependent on
fire weather, as they can be deliberate, one would expect a difference in the performance of the (PEAT-)FWI between these two
seasons, with a lower performance in spring. However, this difference is not seen here for FWI$_{ref}$, as it performs approximately
the same for both seasons, indicating that this effect might be negligible. The main difference in performance for PEAT-FWI
can be due to a lower impact of the PEATCLSM input on the fire danger due to more anthropogenic ignitions and is thus less
influenced by the soil hydrology, as discussed earlier in this section.

### 4.2.2 Opportunities for operational FWI products

CDF-matching was used to adapt the moisture codes within the FWI over peatlands. Since this method keeps the integrity of the
calibrated FWI structure, it can be considered a conservative way of adapting the FWI. Other adaptations that fundamentally
change the FWI structure might be more effective and show even greater improvements. However, the conservative nature of
CDF-matching makes the inclusion of hydrological variables more accessible for centers of operational products as well as the
FWI user community that is used to a certain interpretation of the different indices.

In this study, we used hydrological variable estimates based on assimilating SMOS L-band observations into PEATCLSM
simulations. An operational alternative is the soil moisture data assimilation product associated with the SMAP (Entekhabi



et al., 2010) mission. The operational SMAP Level 4 soil moisture product (SMAP L4_SM; Reichle et al., 2019) was recently updated to incorporate the use of PEATCLSM over peatlands (SMAP L4_SM v7; Reichle et al., 2023). With a rather low
latency time of 2.5 days, the peat moisture variables could be implemented in an operational FWI forecast product assuming PEATCLSM variables from SMAP L4_SM v7 are properly extrapolated in time by driving either a PEATCLSM type of modeling approach or an appropriately trained emulator of it with weather forecasts.

## 5  Conclusions

Even though the Canadian Fire Weather Index (FWI) was originally developed for an upland jack pine forest (Van Wagner,
1987), it is used globally across a variety of fire environments (Di Giuseppe et al., 2020; Taylor and Alexander, 2006). However, the applicability of the current FWI is often questioned over certain environments, such as peatlands (Waddington et al., 2012). This study aimed at replacing the original moisture codes of the FWI (Drought Code (DC), Duff Moisture Code (DMC), and Fine Fuel Moisture Code (FFMC)) over peatlands with soil moisture and water table estimates obtained by assimilating SMOS L-band brightness temperature observations into a land surface model with peatland-specific modules. The peat-specific
hydrological estimates were first rescaled to the FWI moisture codes using cumulative density function (CDF) matching to preserve the integrity of the original FWI system. We systematically replaced different moisture codes to evaluate their impact on the performance of estimated fire danger and to find the optimal use of the hydrological variables for a PEAT-FWI.

We evaluated the capacity of these new PEAT-FWI in predicting peat fire occurrence using a "Hits and Misses" analysis and Receiver Operating Characteristics (ROC) curves and compared them against the original FWI ($FWI_{ref}$). For this evaluation,
fire data from the global fire atlas (GFA; Andela et al., 2019) was used. Due to the strong bimodal seasonality of boreal peat fires, this dataset was split into early (before Day of Year (DOY) 130) and late (after DOY 130) fires. For the early-season fires, we hypothesized that the main fuel is dead plant matter from the previous growing season. For the late-season fires, on the other hand, the main fuel was thought to be peat organic material and living vegetation from the current growing season, which is still linked to soil hydrology. The main results are as follows:

1. For the early fires, our results indicate an improvement in FWI performance when adapting only the two deeper soil moisture codes of the original FWI with hydrological variables while further adaptations of the fine fuel moisture code and the elimination of direct wind impacts on FWI clearly deteriorate results.

2. For the late fires, the greatest improvements were found when adapting all original moisture codes, including the fine fuel moisture code, with hydrological variables. Even the impact of wind could be removed without deterioration. This
stands in contrast to the results of the early fires and indicates that late-season fires are more hydrologically driven than early-season fires.

3. A regional evaluation for the late fires shows that the improvements are consistent across different regions (Alaska, Canada, Europe, Siberia).



4. For operational applications, we suggest a varying adaptation of the FWI over peatlands, starting with replacing only
DC and DMC in the beginning of the season and gradually increasing the weight of an adjusted FFMC from early- to
       late-season fires.

Based on our results, we conclude that adapting the FWI with hydrological information is beneficial in estimating peat fire occurrence. However, we emphasize that the FWI is originally not designed to predict fire occurrence, but rather to estimate fire danger. A high FWI value does not necessarily indicate a fire occurrence, nor does a low value make a fire impossible.
However, we assumed that the occurrence of a fire in the GFA indicates a high FWI value. We argue that this is a justified assumption as a fire needs to reach an area of 0.25 km$^2$ to be detected by the MODIS instrument, which is the basis of the GFA. Note that this study focused on assessing the FWI performance by only looking at fire occurrence, but further studies looking at the FWI and burned area could provide valuable additional insights.

In this study, we used a conservative approach for the adaptation of the FWI in the sense that we did not change the original
structure of the FWI but only replaced the different moisture codes after CDF matching the new information to the CDF of the original moisture codes. This ensures that our method could be used operationally without fundamental changes to the original system. With the aid of the SMAP_L4 v7 product (Reichle et al., 2023), similar data to the one used in this study can be routinely downloaded and easily incorporated into an FWI framework.

*Code and data availability.* The MERRA-2 data (Gelaro et al., 2017) were obtained from https://goldsmr4.gesdisc.eosdis.nasa.gov/data/
MERRA2/. In order to access the data, an account at https://urs.earthdata.nasa.gov/home is needed. The PEATCLSM-SMOS dataset is downloaded from https://doi.org/10.5281/zenodo.3731652. The GFWED code used for the FWI calculations can be downloaded from https://portal.nccs.nasa.gov/datashare/GlobalFWI/v2.0/20201013.GFWEDCode.tar.gz. The GFA data used for the evaluation is downloaded from https://doi.org/10.3334/ORNLDAAC/1642. The CWIM3 data can be obtained on request from Canada Centre for Mapping and Earth Observation, Natural Resources Canada. The CDF-matching is done using the pytesmo python package available on https://doi.org/10.5281/
zenodo.7780805.

*Author contributions.* JM developed the framework, performed the experiments, and analyzed the data, supervised by GDL and MB. NA provided the GFA and guided the processing of this dataset. AF processed the GFA into the peat fire dataset. RF provided the code for and provided guidance to the FWI calculations. SV provided topical expertise for the interpretation of results. JM took the lead in writing the manuscript, and all authors helped to shape the research, analysis, and manuscript.

*Competing interests.* The contact author has declared that none of the authors has any competing interests



*Acknowledgements.* J. Mortelmans thanks the Research Foundation - Flanders (FWO) for funding this research (FWO.G095720N). The computer resources and services used in this work were provided by the High Performance Computing system of the Vlaams Supercomputer Center, funded by the Research Foundation - Flanders (FWO) and the Flemish Government. The contribution of S. Veraverbeke was supported by the Dutch Research Council (Vidi grant 016.Vidi.189.070) and European Research Council through a Consolidator grant under
the European Union's Horizon 2020 research and innovation program (grant agreement No. 101000987). We thank Brian Brisco[†], Kevin Murnaghan, and Masoud Mahdianpari for providing the CWIM3 data and valuable insights into the data.





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
