# Peer review of "Improving the Fire Weather Index system for peatlands using peat-specific hydrological input data"

_EGUsphere, 2023_

## Author Comment (AC1)

We very much thank the reviewers for their thorough and constructive reviews on the manuscript. Below we give a detailed response to each comment and we discuss how we used the comments to improve the manuscript.

*The reviewer comments are shown in italic fonts*, our answers are in blue (normal fonts). **Proposed changes in bold.** When referring to figures in this response letter, the prefix 'R' is used before the figure number.

*Reviewer #1 (Sophie Wilkinson):*

We are very grateful to the reviewer for the very thorough review of our paper. The detailed comments were helpful to improve the paper.

*General Comments:*

1. *The authors conduct 4 experiments, where different combinations of moisture codes (FFMC, DMC, DC) in the FWI system are replaced by outputs from the PEATCLSM model (adjusted for L-band assimilation; as per Bechtold et al., 2020). Hydrological model variable outputs are rescaled to match CDFs of moisture codes. The "experimental" variables are then compared against the ability of the original FWI components to predict fire occurrence in peatlands.*

   *The work is a much needed step towards developing fire risk prediction methods for peatland fires however the approach appears to diverge from the original purpose of the FWI system, requiring additional justification and discussion. The FWI was developed to estimate fire danger, whereas here the (PEAT-)FWI is evaluated in its ability to predict fire occurrence. As stated, the two are likely correlated but this is an important note that needs to be addressed more thoroughly before the conclusion.*

We agree that the FWI was not developed to estimate fire occurrence. We originally evaluated the (PEAT-)FWI against 'fire ignition', but changed this to 'fire presence', i.e. looking at all days with a fire. This is in line with the work of Di Giuseppe *et al.* (2020; https://doi.org/10.5194/nhess-20-2365-2020) at ECMWF. Because there are no large-scale 'fire danger' datasets available, we are limited in our evaluation methods. We believe that 'fire presence' is the next best option, because of its high correlation with fire danger. We made it more clear before the conclusion and throughout the paper that a high fire danger does not necessarily indicate a fire and vice versa, due to the importance of the spatio-temporal variability of ignition sources. With fire presence, results are less dependent on the ignition since we expect wildfires to persist on average longer for a high fire danger.

Throughout the text, every time we mentioned ignition as our evaluation method, we changed this to 'fire presence'. Figures 7, 8, and 9 are replaced by the following figures:

[Figure]

*Figure R1: Replacement of Figure 7. This figure is now based on the evaluation against fire presence, instead of fire ignition alone. Additionally, for this figure, we added the n of early and late fires, as well as how many of these fires are a hit for the reference FWI and of the EXP.*

[Figure]

*Figure R2: Same as Figure R1, but as replacement for Figure 8.*

[Figure]

*Figure R3: Same as FigureR 1, but as replacement for Figure 9.*

2. *The discussion reads more like a further description of the results with some added justification, but no real discussion of the impact on predicting fire danger. E.g., Replacement of FFMC with sfmc results in smoother moisture values for the top fuel layer… is this supported by research of peat fuel moisture? What is the impact on predicting fire danger i.e., hits, misses, TPR, of EXP3 and 4 vs FWI and EXP1 and 2. It might be pertinent to focus on the net change or the misses to hits (green) % change for each EXP, as this is the aim of the study – to better predict fire danger (or in this case, occurrence).*

We agree that it is best to focus on the net change of the different experiments compared to the reference FWI calculation. We made this more concrete and focus more on the relative difference in Figure 8 (Figure R1 in this document), by emphasizing the approximately 20% net improvement instead of the separate 28% improvement and 8% deterioration. We have shifted the discussion to focus more on these net changes.

3. *The title appears to be slightly misaligned from the work. I would suggest something like "Using an adapted Fire Weather Index to predict fire occurrence in Boreal peatlands." Or "Modifying the Fire Weather Index System for peatland wildfire occurrence using hydrological modelling".*

We agree that a different title might be more appropriate. Based on the comment of Reviewer 2, we think a title without the microwave aspect might be more suitable. We like your second suggestion and will use that as a title in a slightly different way to subtly include the satellite aspect (it is not purely modeled input that we used):

*Improving the Fire Weather Index System for peatlands using peat-specific hydrological input data*

*Methodological Comments:*

4. *Can you provide justification for your assumption that "soil moisture doesn't control ignitions in early season"?*

It is important to note that we are not suggesting that soil moisture has no influence at all on ignitions during the early season. Rather, our point is that other variables may become more important in determining the presence of fires at this time of the year.

This perspective is supported by the findings of Parisien *et al.* (2023), who have reported that human-caused fires are prevalent during the spring season. These fires tend to occur less frequently within the organic layers of the soil. Consequently, it is often unnecessary for the deeper soil layers to be particularly dry to facilitate ignitions during the early season (Parisien *et al.*, 2023). Instead, human ignitions are mostly influenced by the availability of dry surface fuels, such as leaf litter, which are typically accessible before understory plants begin to green-up and trees leaf-out (De Groot *et al.*, 2013).

The dead surface fuel that is available, is only loosely connected to soil moisture levels. Additionally, the still-living biomass, including tree branches, typically maintains low moisture content at the start of the growing season, before leaf-out, especially when the soil has either not or only recently thawed (De Groot *et al.*, 2013).

We include the following in our revised manuscript (lines 70-74 of the old manuscript):

**… De Groot *et al.* (2013) and Parisien *et al*. (2023) reported that anthropogenic ignitions peak during spring (early season) and are primarily influenced by dead surface fuels, such as leaf litter, and living vegetation that may be still dry from the cold winter season, e.g. due to a ground that only recently thawed. These dead or dry surface fuels have a weak connection to soil moisture (De Groot *et al.*, 2013; Parisien *et al*., 2023). Furthermore, Parisien *et al*. (2023) noted that late-season fires are predominantly ignited by lightning, typically igniting in the organic soil layers further emphasizing the critical role of soil moisture (Parisien *et al*., 2023).**

5. *What is the principle from Li et al., 2010, briefly?*

Thanks for catching an error. After checking the used Python package (*pytesmo*) and the paper of Li *et al.*, (2010), we noticed that the principle from Li *et al.* (2010), which was about bias correcting future predictions, would not be applied in our study. Since our study does not involve future predictions, the principle of Li *et al.* is not used. Instead, we in fact only used the default CDF-matching approach that is provided within the *pytesmo* package.

We changed lines 140-141 to:

**Instead, to maintain the integrity of the original FWI, we matched the temporal Cumulative Density Functions (CDFs) of the PEATCLSM output variables to those of the corresponding moisture codes for each grid cell using the Python pytesmo package (Paulik *et al*., 2023).**

> 6.   Can you clarify how matching CDFs per grid cell removes dry biases over the Boreal Plains? Li et al. states: "the dependence of the bias correction results on the choice of time period for training" hence, how does your training period affect BP projections?

Yes, the bias correction strongly depends on the time period of the training data, which is in our case 2010-2018. If the Boreal Plains (BP) was unusually dry/wet in this period, that will affect our CDF matching. However, the mentioned dry biases refer to the model bias of PEATCLSM. PEATCLSM, as stated by Bechtold *et al*. (2019), has a possible dry bias over the BP. We will clarify this in the revised version of the manuscript to avoid misunderstandings.

We rewrote the paragraph starting at line 137 of the old manuscript:

**By performing this CDF-matching on a per-grid-cell basis, any spatial biases present in the PEATCLSM output, such as a possible dry bias over the Boreal Plains in Canada (Bechtold *et al*., 2019), were removed. It is worth noting that any spatial bias in (the moisture codes of) the FWI will thus be maintained in the new PEAT-FWI with this per-grid-cell approach. However, our primary objective is to underscore the value of temporal peatland hydrological data within an established fire danger rating system.**

> 7.   What is the potential impact of using MERRA data to calculate the (original) FWI? This is not how the FWI system is used operationally (interpolated weather station data is most common), and therefore you may be reducing the quality of the FWI output before you start. For example, limited ability to represent low RH has been found in some global climate models and Field et al. discuss issues with MERRA's accounting of precipitation.

Indeed, it is important to acknowledge the potential impact of using MERRA-2 data to calculate the original FWI. The use of reanalysis data, like MERRA-2, can introduce biases in comparison to real observations obtained from weather stations.

Additionally, we must consider that MERRA-2 data serves as the meteorological forcing for the PEATCLSM simulations. The key point here is that both the original FWI and PEAT-FWI, which incorporates PEATCLSM data, share the same input data source: MERRA-2. Consequently, any issues or biases within the MERRA-2 dataset, such as its representation of precipitation or relative humidity, are carried through into both the FWI and PEAT-FWI.

This approach ensures a level playing field for the evaluation of both systems, as they operate with identical data, including any potential biases. It is worth noticing that while MERRA-2 may have limitations in representing specific meteorological variables, these limitations propagate to both the original FWI and PEAT-FWI. This consistency in data sources facilitates a fair and balanced evaluation of the two systems.

By taking this into consideration, we can more accurately assess the specific contributions of the PEATCLSM dataset and the CDF-matching approach in enhancing the performance of the PEAT-FWI compared to the original FWI.

We included the above discussion in the revised version. See also the response to comment 25 for the added paragraph on MERRA-2.

8. *Can you justify the use of fire ignitions as a measure of fire danger? Or caveat the need for this simplification/assumption via the difficulty of modelling fire danger (intensity) for peatlands.*

Based on your comment and a comment of Reviewer 2, we moved away from using purely 'fire ignition' to 'fire presence', encompassing all days with recorded fires. This way, the evaluation will depend less on the spatio-temporal dynamics of the ignition sources. The new results are presented in Figures R1-R3 of this document.

We are aware that the FWI was not primarily designed for predicting fire occurrence. However, due to the lack of real 'fire danger' datasets, we are somewhat constrained in our evaluation methods. We consider 'fire presence' as the most viable alternative due to its strong correlation with fire danger and provide this justification in the revised version.

9. *What is the historic period for evaluating FWI? 30 years?*

In this study, with "historic", we mean the data we have available, which is from 2010-2018. We included this in the revised version:

**This threshold was chosen as the 90th percentile of the historical (2010-2018) FWI values for the late season, similar to Di Giuseppe *et al.* (2020), and as the 70th percentile of the historical FWI values for the early season.**

10. *How critical are those subjective thresholds of 70 and 90th percentile? Could a simple uncertainty analysis be conducted?*

The choice of the thresholds undoubtedly influence the results. That is part of the reason why we also look at the ROC curves, which do not use an arbitrarily chosen threshold. You could see it as an uncertainty analysis as the ROC curves cover the full range of thresholds. As far as we know, an uncertainty analysis would give the same results as the ROC curve that we currently have, and would not provide additional insights.

In the revised version, we acknowledge the subjectivity of the thresholds and use this as motivation to introduce the ROC curve and its AUC metric.

**Since these thresholds are somewhat arbitrarily chosen and the value of the thresholds influences the results presented in this study, the next section (Section 2.2.4) discusses another evaluation method that uses the full range of possible thresholds.**

> 11. *It might be pertinent to focus on the net change or the misses to hits (green) % change for each EXP, as this is the aim of the study – to better predict fire danger (or in this case, occurrence; see above comment).*

We agree to this comment and changed the perspective of the discussion to focus more on the net changes.

> 12. *What proportion of your study area is underlain by permafrost? How might that impact fire danger, and the hydrologic modelling (and therefore PEAT-FWI)?*

We are aware that the presence of permafrost and related hydrological processes will influence fire danger, and the way these processes are modeled will influence the results in this study. PEATCLSM, the land surface model used in this study, simulates freeze-thaw processes and regulates runoff based on the modeled ice fraction in different ice layers, i.e. simulates the most basic permafrost processes to some degree. However, the occurrence of discontinuous permafrost at subgrid scale is not simulated by PEATCLSM, most likely reducing the accuracy of the model over those areas.

Based on the temperature of the deepest modeled soil layer (3 - 13 m), we derived an estimate of permafrost in our study area (see Figure R4). Based on this estimate, approximately 0.90% of the peatlands in our study area are underlain by permafrost, almost all located in northern Canada. On the other hand, the Brown *et al.* (1997) permafrost boundary, shown in green on Figure R4, shows a slightly larger area of peatlands underlain by permafrost. PEATCLSM performance might be worse in those areas due to underestimation of permafrost effects. Overall, we think the vast majority of wildfires in our domain is not in areas of continuous permafrost. We included a paragraph on the role of permafrost in the discussion of revised version:

**Besides these possible shortcomings of the used datasets, the used land surface model, PEATCLSM, has also its limitations. At high latitudes, peatlands can be underlain by permafrost. Fires can cause permafrost to thaw by deepening the active layer (Gibson *et al.*, 2018). PEATCLSM simulates freeze-thaw processes and regulates runoff processes to some degree (Bechtold *et al.*, 2020). However, the occurrence of discontinuous permafrost at the subgrid scale is not simulated by PEATCLSM, possibly reducing the accuracy of the model over those areas.**

**Continuous permafrost occurrence**

[Figure]

*Figure R4: Continuous permafrost occurrence in the peatland pixels of the boreal zone according to the PEATCLSM model (blue pixels) and Brown et al. (1997; green line). Grey shadings indicate peatlands not underlain by continuous permafrost.*

13. What is the total number of peat fire ignitions in the dataset?

We added the following sentences to the paragraph on lines 239-247:

**In total, over the full study area and period (2010-2018), there are 12 321 ignitions on peatlands, resulting in 9025 early and 47 684 late fire presences, i.e. a day with an active fire at a specific location, so in total there are 56 709 active fire presences (at daily time resolution) considered in this study.**

*Line Edits:*

14. Line 17: References could be updated/expanded.

We added extra references:

**Under wet conditions, peatlands act as a sink for atmospheric carbon (Gallego-Sala *et al.*, 2018) but due to natural (e.g. climate change or lightning-ignited wildfires) and anthropogenic (e.g. drainage for agriculture and forestry) disturbances, this sink can turn into a source (Davies *et al.*, 2013; Granath *et al.*, 2016; Loisel *et al.*, 2021; Turetsky *et al.*, 2002, 2004, 2010; Wieder *et al.*, 2009; Wilkinson *et al.*, 2018).**

15. Line 18: Consider "saturated" over water-logged.

Changed

16. Line 62-65: Incorporate this into a paragraph.

Changed

17. Line 70: "The fuel for early-season fires is mostly the dead plant matter of the previous growing season and thus less influenced by current soil moisture" Is this statement supported by literature? Although it is hypothesized (and there is some evidence) that peatland fire severity worsens throughout the fire season, there is much evidence that high peat burn severity is controlled on a much smaller scale (peat properties) and across scales (topographic position, hydrogeological setting), and that extensive peat organic matter can combust during early season fires.

We agree that at a smaller scale, other landscape or peat properties, such as peatland connectivity, distance to the edge of the peatland, fuel type, and whether it is a treed or open peatland, control fire occurrence and dynamics. However, in this study we look at large-scale fire dynamics and controlling factors. We are mainly looking at the average peatland fire behavior, for which we believe this hypothesis holds (see also our response to comment 4). It would be interesting to look at a smaller scale, e.g. individual peatlands, and combine multiple datasets of different peat and topographic properties to describe peatland fire dynamics in more detail in a future study. We included these small scale factors in the discussion of the revised manuscript.

18. Line 70-75: Parisien et al., 2023 refers to boreal-wide ignitions and I'm not sure how context on anthropogenic vs lightning ignitions supports the assumption that soil moisture is having

*different levels of control over wildfire ignitions in early or late season. Note that I don't disagree with this sentiment, just the supporting evidence.*

See our response to comment 4.

19. *Line 76: "Furthermore, the fuel moisture of the latter depends on the peat moisture status (Harris, 2008)". What does this sentence mean? Seasonal change or peatland type or something other?*

We meant that vegetation moisture content depends on soil moisture content. We changed it to:

**For late-season fires, the majority of the fuel in peatland fires is expected to be peat organic material and to a smaller extent living vegetation (Davies *et al*., 2016). While the impact of peatland hydrological conditions on peat fuel moisture is trivial, the moisture content in living vegetation also depends on peat moisture content (Harris, 2008). We, therefore, hypothesize that the replacement of the FWI moisture codes will have different effects on estimating early and late-season fire danger.**

20. *Line 100: "if the fine fuel is dry" is redundant since the fuel moisture is accounted for in the moisture index.*

We removed it.

21. *Line 100: Please clarify what is meant by "without taking further vertical drying into account".*

We meant the drying of deeper soil layers due to the fire on the surface. Van Wagner (1987) state about the Initial Spread Index:

*"the name refers not so much to the behavior of a fire during its early life just after ignition, but rather to the basic rate at which a fire will spread when the fine fuel is dry but further drying in depth is not well advanced."*

That is what is meant with the "vertical drying" in this phrase. We suggest to make this more clear by changing the sentence:

**… without taking further drying in depth, i.e. the drying of deeper soil layers due to the fire on the surface, into account..**

22. *Line 103: I think this can be more concisely written as the "potential" fire danger i.e., if a fire were to ignite, or is already ignited.*

We agree and changed the sentence to be more concise:

**The FWI is a measure of the potential fire danger, i.e. if a fire were to be ignited, or is already ignited, with higher values representing a higher danger.**

23. *Line 106: and historical data and context.*

Agree and changed:

**Instead, users have to define such thresholds themselves, based on expert knowledge of the local environment combined with historical data and context (Van Wagner, 1987).**

> 24. Line 108+: Consider "original" rather than weather-based or actual, FWI calculations.

We changed "weather-based" to "original" everywhere in the text.

> 25. Line 108+: Please provide specifications of the MERRA data – resolution, downscaling applied, etc. Similarly, please provide details on validation or accuracy measures conducted on the GFWED method. Somewhere here it should also be noted that most operational users of the FWI system use specific local weather (met) stations and interpolated station data to calculate FWI.

We acknowledge that local operational users may use data from nearby weather stations in their FWI calculations. This practice is relatively straight-forward with the necessary weather data. However, the accuracy of the FWI calculations depends on the density of weather stations in the area. To know whether the proposed usage of hydrological information from PEATCLSM in the FWI performs better than the weather station data, should be tested by fire managers and their available datasets. We will include this point in the discussion of the revised paper (see also our response to comment 7).

 We included the following information on the MERRA-2 data:

**In this study, the meteorological data needed to calculate the original FWI calculations were taken from the NASA Modern-Era Retrospective Analysis for Research Applications version 2 (MERRA-2; Gelaro et al., 2017). MERRA-2 offers atmospheric data with a spatial resolution of 0.5° x 0.625°, spanning 72 vertical levels, and a temporal resolution of 1 hour. It has been applied without any downscaling. The input meteorological variables for FWI, i.e. $RH_{2m}$, $T_{2m}$, and $V_{10m}$, were based on the instantaneous values at noon of each day. For P, the in situ gauge-corrected accumulated total precipitation of the last 24 hours was used.**

**The actual FWI calculations were conducted using the source code of the Global Fire WEather Database (GFWED; Field et al., 2015), which relies on MERRA-2 as input weather data in the retrospective mode. However, it is worth noting that our study did not entail specific validation of the MERRA-2 based calculations against in situ weather station observations. A comprehensive validation of the GFWED code can be found in Field et al. (2015), which highlights potential biases, particularly for the lower latitudes. For the boreal region, the MERRA-based FWI calculations showed some overestimation of DC, but this depends on the specific region. We are aware that the performance of the FWI is dependent on the quality of the meteorological input data, and in the case of the PEAT-FWI also the hydrological input data which in our case is generated using the same source of meteorological input data for the sake of comparability between FWI and PEAT-FWI.**

> 26. Line 126: Please define DA.

Added. It stands for "data assimilation".

27. *Line 150: fire presence, is that referring to an ignition or an active fire? Or a burned area?*

This is referring to an active fire. We changed the whole focus of this paper to active fire presence, which makes this more clear.

28. *Line 203: Could you comment on the role of human vs lightning caused fires altering the FWI threshold?*

Certainly, in the context of early-season fires, human-caused ignitions, and the resulting fires, are influenced by specific conditions. These fires may occur with a lower FWI due to factors like the initial dryness of the vegetation due to cold air drying tree branches when soil is still partly frozen (Groot *et al.*, 2013). Furthermore, humans may ignite surface fires on peatlands for management practices when peatlands are still wet, i.e. the FWI values are still lower. These specific conditions for human ignitions create a different setting for the interpretation of the FWI compared to lightning fires, which dominate late season fires.

We added this to the revised manuscript:

**This threshold is lower for the early fires, as these fires may occur with a lower FWI due to factors like the presence of dead vegetation of the previous year or the early-season dryness of the living vegetation when soil is still partly frozen (Groot *et al.*, 2013). Furthermore, humans may ignite surface fires on peatlands for management practices when peatlands are still wet, i.e. the FWI values are still lower. These specific conditions for human ignitions create a different setting for the interpretation of the FWI compared to lightning fires, which dominate late-season fires.**

29. *Line 210: Is FWIEXP == PEAT-FWI? Please keep acronyms and notations consistent throughout.*

Changed.

30. *Figure 4: Would be fine as supplemental figure.*

We agree and moved it to the appendix of the manuscript.

31. *Line 253: Despite accounting for X (i.e., relatively small) percentage of ignitions.*

Added:

**In fact, ~90% of the burned peat area from 2010 through 2018 is caused by fires ≥ 2 km$^2$, despite accounting for only 20% of ignitions.**

32. *Figure 6: Can you show which fires are hit by all FWIs and missed by all FWIs? Or give summary stats of these?*

We changed the representation of hits and misses in this figure. In the new figure below (Figure R5), we added the thresholds of the different (PEAT-)FWI as horizontal, dashed lines. This makes it easier

to interpret. If there is an active fire (presented by the black lines and grey shading), a specific (PEAT-)FWI gives a hit if the FWI is above the threshold, and a miss when it is below the threshold. In this case, the first active fire is fully "predicted", i.e. a hit, by all EXP, but not by the reference FWI. The next 4 fires are "predicted" by all EXP and the reference FWI. The second to last fire presence is only a hit for EXP3 and EXP4 and the last fire presence is missed by all EXP and the reference FWI.

We included some statistics on the hits and misses in Table R1. We do not think this would add valuable information to the paper, as this information can already be derived from Figure 7 (Figure R1 in this document). To make it more clear, we added the total number of early and late fires in the caption of that figure, as well as the percentage of these fires that are hits for the reference FWI, i.e. the Hits in Table R1 divided by the total number of fires. The number of hits for the different experiments is then also shown in the figure based on the percentages of hits to misses and misses to hits.

Table R1: information on the number of hits, i.e. detected fires, misses to hits, and hits to misses for the early and late fire season for each experiment.

|  | Hits | | Misses to hits | | Hits to misses | |
|---|---|---|---|---|---|---|
|  | Early | Late | Early | Late | Early | Late |
| Reference | 784 | 13116 | / | / | / | / |
| EXP1 | 837 | 13326 | 60 | 378 | 7 | 168 |
| EXP2 | 812 | 14506 | 101 | 2451 | 73 | 1061 |
| EXP3 | 169 | 21812 | 41 | 12160 | 656 | 3464 |
| EXP4 | 326 | 22532 | 89 | 13170 | 547 | 3754 |

[Figure]

*Figure R5: The same as Figure 6, but with additional information of all fires. T90,EXP indicates the 90th percentile threshold for the respective (PEAT-)FWI used to determine the hits and misses.*

33. *Figure 7: What about which fires are hits to misses and which are misses to hits?*
    *Can you give numbers for the net impact of each EXP?*

Yes. For the net impact of each EXP, one could take the difference between the percentage for the misses to hits and that of the hits to misses. The results are shown in Table R2. We will focus the discussion more on these net changes instead of looking at the improvements and deteriorations separately. We discuss these net changes in text, but will not include Table R2 in the paper, to keep the number of figures and tables limited:

When subtracting the deterioration from the improvement, one can evaluate the net effect of the PEAT-FWI for the different experiments. This shows that EXP1 shows a small net improvement compared to the FWI$_{ref}$ for both the early and late fires of 0.58% and 0.44%, respectively, i.e.

**0.58% of the fires in the early season are better predicted with EXP1 than with FWI$_{ref}$ using the 70th percentile threshold. EXP2 also shows a net improvement for both seasons (0.31% and 2.91% for the early and late season, respectively). For the early season, EXP3 and EXP4 show a net deterioration of 6.82% and 5.07%, respectively. For the late season, both experiments show a large improvement of 18.24% and 19.79% for EXP3 and EXP4, respectively.**

Table R2: Net change of the hits and misses evaluation for each EXP. Positive numbers indicate a net improvement. Negative numbers a net deterioration.

|  | Early fires | Late fires |
|---|---|---|
| EXP1 | 0.58% | 0.44% |
| EXP2 | 0.31% | 2.91% |
| EXP3 | -6.82% | 18.24% |
| EXP4 | -5.07% | 19.79% |

34. *Figure 8: I'd like to see FRP and TPR in a table.*

We believe a table with this information would duplicate the information given in the ROC curves. For the 70th and 90th percentile, the FPR and TPR can be approximately read from the ROC curves, as this is indicated on the ROC curves with a star.

*Discussion:*

35. *Line 325 - 333: I don't find the following analysis useful to the discussion as study regions/ ecosystems are not comparable. Perhaps replace with discussion on the challenges of mapping both peatlands and fires and then discuss why your approach is well-suited.*

We dropped the discussion of number of fires and burned areas with other studies. We did not replace this discussion with a discussion on the challenges of mapping fires and peatlands, as we feel that is already covered in the next paragraph dedicated to the challenges of the GFA and the PEATMAP.

36. *Line 338: Merge this with above paragraph.*

Done

37. *Line 361: missing bracket*

Added

38. *Line 361+: Showing data for one time period for one specific geographical location is helpful for readers to understand the differences in temporal dynamics of the FWI and PEAT-FWI however it doesn't stand up when discussing the overall efficacy of the PEAT-FWI, in particular because fire ignitions are so stochastic and therefore you could just as easily show a time/place where high PEAT-FWI does not co-occur with fire occurrence. Specifically, the wording: "As these two values reach their maximum, five fires are observed, indicating that they estimate fire danger relatively well." is not justified in this context.*
39. *Line 361+: Further, discuss the role of lightning/ignition in fire occurrence.*

We rewrote the discussion keeping this in mind. We emphasized that this part of the discussion is indeed based on one fire season of one specific 9x9 km$^2$ pixel. This pixel is randomly chosen, but indeed, we could arbitrarily pick a really good/bad pixel to show. We softened the conclusions made from this figure, but we still think it shows valuable information and it is worth including this figure and the discussion.

**However, it is important to keep in mind that this time series, and thus the conclusions one can draw from it, might not be generally applicable to the whole study area and period. A different location or a different fire season might result in different conclusions and a different performance of the PEAT-FWI compared to FWI$_{ref}$.**

*40. Line 373 – 385: Some of this could go in the Methods; description of FWI moisture codes.*

We agree that some of this part of the general description of the temporal dynamics of the FWI components, and thus could be part of the Methods section. However, we believe that it helps to understand the figures. By moving this description completely to the Methods section, it would become more difficult to follow the discussion of the time series. However, we added, in short, a description of the temporal dynamics of the FWI components to the Methods:

**… The FFMC represents a fast-drying layer. Being the top layer of the FWI structure, the FFMC is most reactive to rainfall, relative humidity, wind, temperature, drying, and smaller rainfall events as compared to the other moisture components. Van Wagner (1987) represented this sensitivity to drying as the time lag, i.e. the time necessary to lose approximately two-thirds of the free moisture above equilibrium. For the FFMC, this time lag is two-thirds of a day, showing a fast rewetting and drying after a rainfall event, resulting in large day-to-day variations.**

**… The time lag of the DMC is 12 days, resulting in a much slower response to rainfall events as compared to the FFMC, and less day-to-day variation.**

**… The time lag for the DC is 52 days, showing even less day-to-day variation than the DMC.**

For the part in the Discussion, we changed it to:

**Contrary to the DC and DMC, the FFMC of FWI$_{ref}$ shows significant day-to-day variability. This divergence can be attributed to the design of these moisture codes. As outlined by Van Wagner (1987), the FFMC is designed to represent moisture content in the litter layer, which is characterized as relatively fast-drying and high sensitivity to small rainfall events. The DMC and DC, on the other hand, represent the upper 5-10~cm and 10-20~cm layers of the soil (De Groot, 1987) and are far less responsive to short-term fluctuations in rainfall. This is due to their more extensive moisture storage capacity.**

**The swift response of the FFMC to rainfall can be attributed to the fact that it is the topmost layer of the system, intercepting precipitation before it penetrates deeper layers. Moreover, the FFMC's top layer position renders it more sensitive to various environmental factors, such as relative humidity, wind, temperature, and drying conditions. To quantify this difference in sensitivity to drying, Van Wagner (1987) introduced the concept of time lag, signifying the duration required to lose approximately two-thirds of the free moisture above equilibrium. They defined the time lag**

**for FFMC as two-thirds of a day, for the DMC it is 12 days, and for the DC it extends to 52 days (Van Wagner, 1987). This distinction underscores that FFMC dries out much more rapidly and subsequently experiences faster increases after rainfall events. These dynamics contribute to the pronounced day-to-day fluctuations evident in Figure 5.**

41. *Line 384/6: Replace "spiky" with "sensitive" or "highly fluctuating"*

Changed

42. *Line 386 – 394: Comment on the potential for hydrologic disconnection of water table and peat surface under deep water table conditions (and/or frozen ground).*

Very true, there is a potential of disconnection in extreme circumstances, such as extreme droughts. We will clarify that this can also potentially happen in the PEATCLSM simulations, however, it barely occurs when applying a parameter set representing natural (undrained) peatlands.

We added the following sentence to include this aspect in the paper:

**… By replacing the FFMC with PEATCLSM sfmc, a strong vertical soil moisture coupling from the PEATCLSM hydrological model is introduced into the FWI system, resulting in a smoother FFMC. However, it is important to note that in extreme circumstances, such as a prolonged drought, a disconnection between the water table and the peat surface can occur which was, however, barely indicated by simulation results of PEATCLSM that are based on a parameter set representing natural (=undrained) peatlands.**

43. *Line 425: "The fact that EXP3 mainly shows misses to hits and even fewer hits to misses than EXP1and EXP2 for the early fires indicates that these early fires are not so much driven by hydrology." Is this the correct way around? I see many more red bars (hits to misses) for EXP3 in early fires. In general I find this terminology confusing and wonder if there's a more concise way to describe it.*

That was indeed the other way around. We corrected the sentence and made the wording a bit more concise:

**The fact that EXP3 mainly shows deteriorations and even fewer improvements than EXP1 and EXP2 for the early fires indicates that these early fires are not so much driven by hydrology.**

44. *Line 444: "Overall, the PEAT-FWI performs worst over Canada, which could be related to the more aggressive fire management compared to e.g. Siberia, which influences the fire behavior overall. It can cause fires with a very high PEAT-FWI to be extinguished before being detected by satellite remote sensing, eventually lowering the performance of the PEAT-FWI." Do you have evidence of this occurring?*

We rephrased this to make clear that this is one possible explanation for why the (PEAT-)FWI differs between these geographic regions. With the new results, based on fire presence instead of solely on ignition, Alaska and Canada perform similar but still worse than Siberia. This hypothesis is based on

the work of Flannigan *et al.* (2009; https://doi.org/10.1111/j.1365-2486.2008.01660.x) and Kharuk *et al.* (2021; https://doi.org/10.1007/s13280-020-01490-x).

We changed it to the following:

**Overall, the PEAT-FWI performs worst over Alaska and Canada. One possible explanation could be a more aggressive fire management in North America compared to e.g. Siberia (Flannigan et al., 2009; Kharuk et al., 2021), which influences the fire behavior overall. While the fire suppression system in Russia was largely successful in the early 1990s, with the fall of the Soviet Union, budgets got reduced and the Russian fire suppression system became less effective (Flannigan et al., 2009; Kharuk et al., 2021). While the Russian fire suppression system has been under redevelopment since the early 2000s, it has not yet reached its former efficiency levels (Kharuk et al., 2021). A more aggressive fire management can cause fires with a very high PEAT-FWI to be extinguished before being detected by satellite remote sensing, eventually lowering the performance of the PEAT-FWI.**

45. *Line 449: "They also found that the Canadian fire season peaks earlier than the Russian fire season (April-May for Canada versus May-June for Russia; De Groot et al., 2013)". After reviewing the paper I do not agree with the above statement, see Table 7 for monthly fire stats. See Hanes et al 2019 and Parisien et al., 2023 for more up-to-date analysis of the Canadian fire season. Note its own binomial distribution (which does not align with the GFA), and the peak in July.*

Indeed, it seems that we confused the peak of fire weather with the peak of fire occurrence in the study of De Groot *et al*. (2013). Based on their Table 7, the fire occurrence in Canada peaks in July and in May for Siberia. Based on Table S1 of Parisien *et al*. (2023), the median DOY of the wildfire count in Canada is DOY 143, or approximately May 23.

Based on the data of the GFA, fire ignitions peak in July in Canada, which is a bit later than the peak as stated by Parisien *et al*. (2023). In Siberia, there are two peaks according to the GFA, one in April and one in July. This does not really align with the fire seasons as discussed by De Groot *et al*. (2013). However, De Groot *et al.* (2013) had 10.247 fires in their Russian study area that were not dated, so these fires can either strengthen/lengthen the peak in May that they found, or create a new peak later in summer.

We rephrased the statement in our paper to the following:

**They also found that the Canadian fire season peaks later than the Russian fire season (July for Canada versus May for Russia; De Groot et al., 2013). More recent studies showed that the Canadian fire season peaks more at the end of May, possibly indicating a shift in the fire season over the last years (Parisien et al., 2023). The GFA data showed that the main peak in fire occurrences in Canada occurred in July, aligning with the data of De Groot et al. (2013). For Siberia, the GFA showed two equal peaks in fire occurrence, one in April and one in July. These peaks do not align with the findings of De Groot et al. (2013). However, De Groot et al. (2013) noted that 10~247 fires in their Russian study area could not be dated. These fires could either strengthen or lengthen the peak they found in May or create a new peak later in summer.**

**The difference between the Canadian and Siberian fire seasons could explain part of the difference seen here between these regions.**

---

## Author Comment (AC2)

We very much thank the reviewers for their thorough and constructive reviews on the manuscript. Below we give a detailed response to each comment and we explain how we used the comments to improve the manuscript.

*The reviewer comments are shown in italic fonts*, our answers are in blue (normal fonts). **Proposed changes in bold.** When referring to figures in this response letter, the prefix 'R' is used before the figure number.

*Reviewer #2 (Francesca Di Giuseppe):*

We are very grateful to the reviewer for the very thorough review. The comments and proposed suggestions were very helpful to improve the paper.

*The fire weather index improved for boreal Peatland Using Hydrological modelling and satellite based-L band microwave observations*

*By Mortelmans et al.*

*The paper investigates the use of hydrological model outputs specifically designed for peatland to replace part or all components of the FWI danger rating systems. It responds to a known limitation of the system that, while developed for above ground fires, has been, and still is, commonly used in global fire forecast systems.*

*It is well known that the FWI does not correlates very well with fire activities where the fuel is very dissimilar to forests. So I believe it is a valuable idea to address this problem. The paper is well written and explores in very much details the source of predictability that can arise by employing the water table and/or the soil moisture content. Certainly I recommend publication of the manuscript as it provides an useful framework to improve fire danger prediction.*

*However given the results that clearly highlight that the direct use of the water table is a better predictor than the FWI for peatland, I am asking the author if suggesting to rescale this variable to match the value of the FWI is really the right thing to do? Is it going in the right direction wanting to retain the infrastructure of the FWI at all costs? The water table is a physical measure that could be measured and even adjusted through the use of satellite observations while the FWI is an empirical transformations of the fire intensity that its calibrated on a specific ecosystems. To CDS-match the FWI seems a weird think to do if you have a metric (the FWI) that is not very correlated to the fire activities in peatland. Even more so as you indead then evaluate against fire activity and not fire danger.*

*The question for me would be why do not directly rescaling the water table by training it to detect actual fire activities ? Along those lines I developed the FOPI which was trained on observed fire activities and did not attempt to rescale the FWI while still using the FWI as a driver for the fire weather component.*

*Importantly, when you train for fire activity, your output is somehow a probability which is more intuitive to understand. Another benefit of using directly the water table would be that when these variables are improved by the model or the assimilation system, this improvement shoud benefit the fire danger indicators in cascade. With the empirical structure of the FWI if you improve the moisture content estimation of the duff layer do you really improve the DMC ?*

*Indeed the motivation to be using the FWI infrastructure is provided in the paper. The FWI is easily interpretable by fire agency. This is a good motivation. Still I am asking the author to elaborate a bit as I think they should discuss what would be the benefit of also shifting toward a more physical based indicators of fire danger.*

We appreciate the thoughts about a possible fundamentally different algorithm for estimating fire danger and/or fire activity in peatlands. We agree shifting to another indicator may have many advantages. However, we also see some important practical disadvantages, especially with regard to the short-term impact of our study. Intentionally, we decided for the very conservative approach of CDF matching for two main reasons:

1) Proof of the positive impact of peat-specific hydrological information: We believe that it is a very transparent and valuable approach to prove the impact of peatland hydrological information by incorporating it in an existing fire danger prediction system. Having this very well known reference system, users may be more easily convinced to adopt the use of peatland hydrological information. In contrast, the performance of a totally different system is much more difficult to specifically analyze for the impact of the new peatland hydrological information. We see such an issue, for example, in the interpretation of the results in Mezbahuddin *et al.* (2023) in which a model with peat hydrological input was shown to perform better than one with weather data only. However, it remained unclear how well their reference system, i.e. a new machine learning setup, was able to extract the information from the weather data. Perhaps the weather-based machine learning model performed even worse than the original FWI? This was not tested in Mezabahuddin *et al*. (2023). In contrast, in our study, we show the value of the peatland hydrological input in an established system, i.e. the value of the peat hydrological information becomes unambiguously clear.

2) Interpretation based on established FWI framework: A completely new approach will complicate the interpretation of the results while the new PEAT-FWI can be interpreted in the same way as the results of the original FWI. With the different experiments presented in the paper, we show that for the early fires, keeping part of the original FWI structure in the system is better than solely relying on the hydrological information.

We have included the following text in Section 4.2.2: Opportunities for operational FWI products and advancing fire danger models

**We employed CDF-matching to adjust select moisture codes within the FWI over peatlands. This conservative approach makes the inclusion of hydrological variables more accessible to operational centers and the user community accustomed to the existing FWI system. It provides a transparent and valuable means to demonstrate the impact of peatland hydrological data.**

**By contrast, using a totally different system would make it difficult to evaluate any performance gain specifically introduced by new peatland hydrological information. For example, Mezbahuddin *et al.* (2023) showed that a machine learning model with peat hydrological input performed better than one purely relying on weather data. However, this study did not offer a comparison with the original FWI system, leaving uncertainty about the true effectiveness of the peatland hydrological data. A study comparing our proposed PEAT-FWI against such a machine learning algorithm could offer new perspectives for future peatland-specific fire danger rating system frameworks.**

*Few more comments:*

1. *I would put figure 4 and all the detailed explanation of the ROC curve derivations in an appendix. I think is quite a standard metric and while is nice to have a refresh in the paper it is distractive to have it in the methids section.*

We agree and moved it to Appendix 1.

2. *I would not mention satellite L-band Microwave Observation in the title. The focus here is not much how the water table is calculated but the use of the Hydrological model. Probably the same results would hold with another model.*

We agree and based on this comment and on a comment from of Reviewer 2, we changed the title into:

***Improving the Fire Weather Index System for peatlands using peat-specific hydrological input data***

3. *How easily available would this prediction of water table be for the forest agency to be able to calculate the new FWI-PEAT Index? Maybe a short discussion of the complexity of creating this index could be provided.*

The hydrological data is freely available to download from https://nsidc.org/data/spl4smgp/versions/7 and has a latency of 2.5 days. We decided on the CDF-matching of the hydrological data to keep the complexity of this index rather low and to allow for an easy interpretation of the results. We will add a short paragraph to the discussion about the operational use of the proposed index on the availability of data and complexity of creating the PEAT-FWI.

4. *You use fire ignition as a fire activity indicator but FWI expresses a measure of fire intensity. I am not sure if this has an impact on your results.*

This definitely has an impact on the results. Based on your comments and those of Reviewer 2, and to be more in line with your previous work (Di Giuseppe *et al.*, 2020; https://doi.org/10.5194/nhess-20-2365-2020), we decided to change from fire ignition to fire presence, i.e. every day that a fire was present is taken into account instead of only the first day. This changed our results for the better. We will replace Figures 7, 8, and 9 with the following figures:

[Figure]

*Figure R1: Replacement of Figure 7. This figure is now based on the evaluation against fire presence, instead of fire ignition alone.*

[Figure]

*Figure R2: Same as Figure R1, but as replacement for Figure 8.*

[Figure]

*Figure R3: Same as FigureR 1, but as replacement for Figure 9.*

5. *I haven't quite worked out why a CDS matching would remove a bias ? Could you give me more datails ?*

The bias we were talking about, is a spatial bias in the PEATCLSM product. PEATCLSM has likely a dry bias e.g. over the Boreal Plains as indicated by field measurements (Bechtold *et al*., 2019). By applying the CDF-matching on the timeseries of each grid cell individually, we remove any possible spatial bias in PEATCLSM data. In other words, the spatial pattern of long-term statistics of PEATCLSM is matched to that of the FWI. This however, assumes that the FWI data are unbiased. If there is any spatial bias in the FWI, this will also be introduced in the new PEAT-FWI. However, as stated earlier, we are targeting in this paper the demonstration of the value of the temporal information in peatland hydrological data in an established fire danger system, ignoring any possible potential of the hydrological information to reduce spatial bias in the original FWI.

To further clarify this in the paper, we changed lines 139-145 of the old manuscript to:

… While this ensures that the CDFs of the PEATCLSM output variables and those of the corresponding FWI moisture codes match, the approach preserves the dynamical features (short-term and long-term anomalies as well as seasonal dynamics) of the PEATCLSM output. By performing this CDF-matching on a per-grid-cell basis, any spatial biases present in the PEATCLSM output, such as a possible dry bias over the Boreal Plains in Canada (Bechtold *et al.*, 2019), were removed. It is worth noting that any spatial bias in (the moisture codes of) the FWI will thus be maintained in the new PEAT-FWI with this per-grid-cell approach. However, our primary objective is to underscore the value of temporal peatland hydrological data within an established fire danger rating system.

6. *Finally I think the discussion is a bit fragmented. For exemple, the limitations of the GFA for peatland are nicely discussed but are not put in the contest of how (or if) they could affect the results presented. Similarly for the discussion about the new index. A clear statement of in how many more cases you are likely to get a good prediction compared to the use of the standard FWI (which you can read from the ROC curves) would certainly benefit the readibility of the conclusions.*

Based on your suggestion and those of Reviewer 2, we revised the discussion, focusing mainly on the net change of the different experiments compared to the reference FWI.

For the net change, i.e. in how many cases we would get a better prediction with PEAT-FWI compared to the reference FWI, we focus on the difference in AUC. Based on the new ROC curves, an improved fire danger prediction is achieved by EXP4 in 6% of the cases for the early fires and in 14% of the cases for the late fires. We added the following part to the discussion:

When subtracting the deterioration from the improvement, one can evaluate the net effect of the PEAT-FWI for the different experiments. This shows that EXP1 shows a small net improvement compared to the FWI$_{ref}$ for both the early and late fires of 0.58% and 0.44%, respectively, i.e. 0.58% of the fires in the early season are better predicted with EXP1 than with FWI$_{ref}$ using the 70th percentile threshold. EXP2 also shows a net improvement for both seasons (0.31% and 2.91% for the early and late season, respectively). For the early season, EXP3 and EXP4 show a net deterioration of 6.82% and 5.07%, respectively. For the late season, both experiments show a large improvement of 18.24% and 19.79% for EXP3 and EXP4, respectively.

---

## Author Response (AR2)

We very much thank the reviewer for the thorough and constructive reviews on the manuscript. Below we give a detailed response to the final comment and we discuss how we used the comments to improve the manuscript.

*The reviewer comments are shown in italic fonts*, our answers are in blue (normal fonts). **Proposed changes in bold.** When referring to figures in this response letter, the prefix 'R' is used before the figure number.

*Reviewer #1 (Sophie Wilkinson):*

We are again very grateful to the reviewer for the very thorough review of our paper. The final comment was very helpful to improve the paper even further.

*Line 85: "While the impact of peatland hydrological conditions on peat fuel moisture is trivial, the moisture content in living vegetation also depends on peat moisture content (Harris, 2008)." This is a new sentence to replace some old wording and I'm unsure of what is being referred to throughout - peat hydrological conditions - is that peatland type (bog, fen, swamp) or other? Inclusion of the word "also" confuses me because the first and second halves of the sentence appear to be arguing in opposite directions.*

We understand the confusion with the current phrasing and have changed it to make more clear what was meant:

For late-season fires, the majority of the fuel in peatland fires is expected to be peat organic material and to a smaller extent living vegetation (Davies et al., 2016). The moisture content of the peat organic material is directly linked to the simulated groundwater level and soil moisture content derived from PEATCLSM simulations. Additionally, peat moisture conditions are known to influence the moisture status of living vegetation as e.g. shown by Harris (2008). Due to the shift towards more living vegetation from early to late-season fires and the related higher importance of peat moisture status both for below and aboveground fuel properties, we hypothesize that the replacement of the various FWI moisture codes will have different effects on estimating early and late-season fire danger.